# Infants make moral character inferences in multi-agent social interactions
Norman J. Zeng ✉, Inderpreet K. Gill & Jessica A. Sommerville

The ability to infer character from behavior is an essential skill that adults use to navigate the social world. We investigated 12- to 24-month-old infants' (Experiment 1, $n = 160$; Experiment 2, $n = 96$) ability to infer an agent's moral character from a complex social situation using a behavioral generalization paradigm that capitalized on infants' visual attention. Infants observed a social event involving an aggressor, victim, and protector or bystander (Experiment 1; Experiment 2 replicated the aggressor and protector conditions). Then, infants saw one of these agents distribute resources fairly (i.e., equally) versus unfairly (i.e., unequally) between two recipients. We found that infants selectively expected protectors and victims to distribute resources fairly and had no significant expectations for bystanders. Infants either expected aggressors to be unfair (Experiment 1) or displayed no significant expectations for aggressors (Experiment 2). Exploratory analyses revealed that infants' moral character inferences were tied to infants' social contact: infants with siblings and daycare experience showed greater moral role differentiation (Experiment 2). These results suggest that infants can make broad character inferences in complex multi-agent social situations, and that their ability to differentiate moral roles strengthens with social experience.

The ability to make dispositional inferences (e.g., kind, intelligent, moral, immoral) is central to navigating complex social environments[1,2], providing a means to predict others' future behavior[3,4]. Although adults spontaneously and unconsciously form a range of dispositional inferences[5,6], inferences about an agent's underlying moral character constitute a central mode through which humans perceive others[7]. Yet, debate exists regarding the developmental emergence of dispositional inferences (in general) and moral character inferences (more specifically), with some scholars suggesting that this ability develops in middle childhood[3], and others arguing that it is present within early childhood[8,9], or perhaps even infancy[10]. Moreover, a critical outstanding question concerns the scope of social situations to which children can apply such moral character inferences, given that social interactions in the real-world often take place in complex, large-scale social environments[11].

Here, we investigate infants' ability to form moral character inferences in the second year of life in the context of complex multi-agent scenarios, in which different agents adopt distinct moral roles that vary along a continuum from morally good to morally bad. Our focus in the current experiments was on broad moral character inferences (e.g., good, bad, moral, immoral), rather than specific moral trait inferences (e.g., helpful and violent), given that research with adults suggests that the former dominates over the latter in impression formation[12].

Research investigating infants' cognitive capacities suggests that infants have the initial building blocks necessary for the formation of moral character inferences. Infants are sophisticated social reasoners: they can distinguish agents from inanimate objects, represent other agents' goals and intentions, and understand that other agents have diverse desires[13–15]. Central to the question of when moral character inferences arise, infants also possess rudimentary moral sensitivities[16,17]. Within the first year of life, infants look and reach preferentially towards helpful characters over hindering characters[18–20]. By the second year of life, infants expect agents to share resources fairly (i.e., equally) between two recipients[21,22], associate fair and unfair actors with praise and admonishment, respectively[23], and preferentially reach towards fair over unfair individuals[24]. Additional work suggests that sensitivities to fairness may emerge in the first year of life[25,26], though scholars have contested whether these studies reflect a sensitivity to fairness or to the social exclusion of a particular agent (given that in these studies the disadvantaged agent does not receive any resources at all)[27].

Recent work has directly addressed the possibility that infants infer moral character by gauging infants' visual reactions to behaviors that are either morally consistent or morally inconsistent with an initial target behavior, a method commonly used to investigate dispositional inferences[4]. Research using this approach suggests that 16- to 18-month-old infants expect that an agent who helps or hinders another agent multiple times will

Department of Psychology, University of Toronto, Toronto, Canada. ✉e-mail: norman.zeng@mail.utoronto.ca

also help or hinder the same agent in future scenarios[28]. Similarly, after watching an aggressor hit and chase a victim before a protector saves the victim from aggression, infants subsequently expect the protector to help the victim over the aggressor and attack the aggressor over the victim[29]. Although these results could suggest an understanding of moral character, they may be more parsimoniously explained either by infants' goal understanding (that the actor has the goal of helping a particular agent), or by infants' understanding of relationships (that one agent likes or views another agent positively)[30]. Thus, to understand whether infants can form character inferences, it is critical to investigate infants' expectations for behaviors enacted towards novel targets.

Three prior studies helped to establish that toddlers can at least make rudimentary moral character inferences. In one study, 25-month-old toddlers suspended baseline expectations that an agent would act fairly after seeing that agent destroy an ingroup member's property (Expts. 1–3) and found it more surprising when this wrongdoing agent subsequently gave most of their resources to an ingroup member versus when the ingroup member gave most of their resources to the wrongdoer[31]. Second, two studies demonstrated that in the second year of life, infants expect agents who previously helped another agent achieve their goal to subsequently distribute resources fairly, whereas infants suspended this fairness expectation for agents that previously hindered another agent[10,32]. Third, another experiment showed that infants aged 14- to 26-months found it surprising when a previously unfair agent helped another agent obtain their goal[32]. When the prior agent acted fairly, however, infants showed no expectations for subsequent helping or hindering behavior.

These prior studies suggest that infants are likely capable of making moral character inferences in dyadic situations for single agents. However, humans live in large social networks and congregate in groups that vary in their sizes[11,33]. Moreover, moral roles not only vary categorically (i.e., prosocial versus antisocial) but are also arranged along a continuum from morally good to morally bad. While some agents within this continuum are easily evaluable, in the case of morally good heroes or evil villains, other agents, such as bystanders or victims, are morally ambiguous and difficult to evaluate[34,35]. To successfully navigate these social environments, individuals must be able to infer moral character for a wide range of agents. Thus, our focus here is on the moral character inferences that infants form for a range of agents that vary in their moral nature and complexity.

In the present study, we investigate infants' ability to make behavioral generalizations to test whether infants can form moral character inferences towards agents in a protective third-party intervention event (Fig. 1)[29]. In this event, an aggressor hits and chases a victim before a protector helps the victim, or a bystander avoids the altercation. Past research demonstrates that

infants prefer victims to aggressors, and protectors to bystanders[29,36,37]. We chose to investigate infants' moral character inferences towards agents in this type of scenario because protective third-party intervention events consist of complex interactions involving three agents, each taking on separate roles.

The agents in this protective third-party intervention event also differ from agents used in past investigations of trait inferences in important ways. In past studies, negatively valanced agents typically engaged in indirect harm towards recipients, such as hindering their goals[10], treating them unfairly[32], or destroying their property[31]. In contrast, in our research, the aggressor engages in direct physical harm, contacting the victim and repeatedly hitting and chasing them. This could lead infants to make stronger moral character inferences, as physical harm is often seen as morally worse than other forms of indirect harm[31,34,38–41].

In addition, infants in our experiment saw two morally ambiguous agents: victims and bystanders. Past work shows that adults can either evaluate the moral character of victims positively (as in the case of virtuous victims[42]), or negatively (as in the case of victim blaming[35]). Evidence also suggests that infants show varied evaluations of victims or victim-like agents. For example, infants preferentially reach towards victims of physical harm over neutral agents[36], suggesting a more positive evaluation of victims than neutral agents. However, they also help agents that possess more versus fewer resources[43] and expect resources to be distributed based on dominance hierarchies, with fewer resources going to the "victim" or submissive agent[44], suggesting negative evaluations. Bystanders are also ambiguous because, although their actions allow harm to occur, adults generally evaluate harm by omission less negatively than direct harm[45]. Thus, testing infants' expectations to a range of moral agents enables us to ascertain whether infants, like adults, view moral behavior as existing on a continuum from good to bad or clearly moral to immoral.

In Experiment 1, we recruited 160 infants ($n = 32$ per condition) aged 12- to 24-month-old. We chose this age range because infants possess the minimum prerequisites for making moral trait inferences by 12 months (i.e., preliminary understanding of agents, fairness, and protective third-party intervention[14,21,29]), and because we were interested in conducting exploratory analyses testing for the presence of age effects. In past studies, age has been an unexplored variable due to low power[32] or restricted age ranges[31], but the presence of age effects or lack thereof can provide information regarding the source of infants' expectations, and whether such expectations are predominantly innate or acquired across development.

To address our research question, infants participated in a violation-of-expectation paradigm. Infants were familiarized with either a protective third-party intervention event or a random-movement control video. Then,

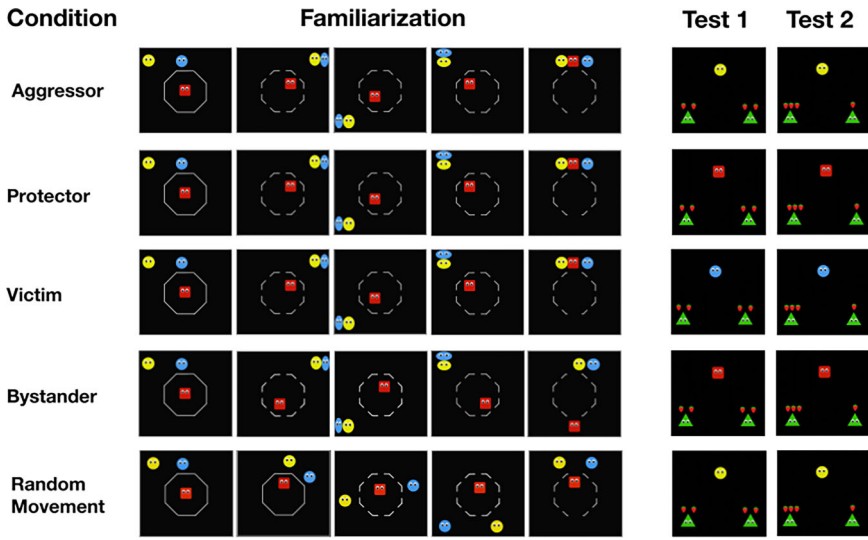

**Fig. 1 | Stimuli and procedure.** Test trials are counterbalanced for order. In the random movement condition, infants saw one of three possible pairs of test videos.

they saw test trials in which one of the prior agents (which differed depending on condition) subsequently abided by distributive fairness norms, sharing four strawberries equally (2:2) between two recipients, or transgressed distributive fairness norms, sharing the four strawberries unequally (3:1). Our primary measure of interest was infants' looking time towards each of the test trials, which we used as an index of expectancy[46].

If infants form moral character inferences in these scenarios, their distributive fairness expectations should differ based on the condition to which they are assigned (aggressor, protector, bystander, victim, and random-movement). More precisely, we predicted that infants would form expectations that aggressors would behave unfairly rather than fairly, and that protectors would behave fairly rather than unfairly. This would be consistent with adult studies demonstrating similar patterns[47] and with work showing that the domains of harm and fairness are closely linked[48]. We did not have directional predictions for the victim or the bystander due to the ambiguous nature of their moral leanings[35,42,45].

In Experiment 2, we conducted a replication of two key conditions of Experiment 1, the protector and the aggressor conditions, to improve the robustness of our results, given concerns over the replicability of psychological studies and of studies in socio-moral development more generally[49,50]. These two conditions were chosen because they have clear directional predictions and because they are the least morally ambiguous roles.

Additionally, we aimed to extend our findings by examining the effect of sibling and daycare experience on infants' moral character inferences. Research with infants and toddlers suggests that infants' degree of social contact with others relates to various aspects of their social and moral cognition[51–53]. For example, in one study, 12- to 15-month-old infants with at least one sibling showed stronger expectations for fair resource distributions than infants without a sibling, perhaps because having siblings allowed infants to gain more first-person experience with resource distributions, boosting their learning about fairness[53]. Based on this research, we expect that infants with siblings and daycare experience will be more likely to make moral character inferences, as these infants should have increased social experience, giving them more opportunities to learn about others' moral character.

## Methods

Both Experiment 1 and Experiment 2 were preregistered (Experiment 1 was preregistered on September 13, 2021: https://aspredicted.org/CN8_5SD, Experiment 2 was preregistered on July 16, 2025: https://aspredicted.org/yi9u4y.pdf). Informed consent was provided by children's legal guardians, and all relevant ethical regulations were followed. Ethics approval was obtained from the Social Sciences and Humanities Research Ethics Board at the University of Toronto. Participants were compensated with a $5 Amazon or Indigo gift card. Data distribution was assumed to be normal, but this was not formally tested.

### Experiment 1

**Participants**. We recruited 160 infants aged 12- to 24-months ($M_{age}$ = 17 months 13 days, $SD$ = 3 months 16 days, Male = 76, Female = 84; parent-reported) from a participant database at a large North American university. We preregistered our sample size based on the sample sizes of prior work with similar methodologies (e.g., per condition: $n = 28$[32]; $n = 32$[31]; our study, $n = 32$), and additional power sensitivity analyses demonstrated that our sample size was sufficiently powered to detect most of our significant effects (Supplementary Methods 1). Caregivers reported that infants were white ($n = 57$), Mixed ($n = 26$), East Asian ($n = 19$), Southeast Asian ($n = 8$), and South Asian ($n = 10$) or did not report ($n = 38$). Twenty-six additional infants participated but were excluded from data analysis because of parental interference ($n = 6$), technical error ($n = 3$), experimenter error ($n = 2$), or because infants were inattentive or fussy ($n = 15$). Infants were initially randomly assigned to the aggressor and random movement conditions, then to the other three conditions.

**Procedure and stimuli**. All experimental sessions were conducted over Zoom. Infants sat in a quiet room in their home, either on their parents' lap or in a highchair. Before the experiment began, parents were asked to hide the experimenter's video and their own video. For calibration purposes, infants were shown five stars presented sequentially in the corners and center of the screen, accompanied by a sound effect. Parents were instructed not to interfere with their infant and to close their eyes or cover their eyes once the experiment began.

Infants were shown four familiarization trials followed by two test trials, consisting of animated video clips of a protective third-party intervention event (familiarizations) and resource distributions (test trials). Each video clip was accompanied by an attention grabber with an image of a xylophone and a music effect. At the end of each video clip, the screen froze on the final frame of the video, and a tone sounded indicating that the experimenter should begin looking time coding. The trial ended after infants looked away from the screen continuously for 2 s, or after a total of 30 s. There was also a minimum looking time of 0.50 s. If infants did not reach the minimum looking time, the trial was repeated. The procedure and stimuli for all conditions are displayed in Fig. 1 and can also be found here: https://osf.io/gjd3p/?view_only=008e27ce76e64ed3a21a4bea89b668a1.

Given that we are using the violation-of-expectation method, we also acknowledge that there is active debate over the validity of these methods[54–56]. However we believe that our use of the violation-of-expectation paradigm is justified, given that (A) past research demonstrates that perceptual novelty and unexpectedness are dissociable[46], (B) infants' visual attention is often consistent with their behavior[57,58], and (C) infants' looking in our study is unlikely to be subject to lower-level perceptual differences as infants in the aggressor, protector, and victim conditions received identical familiarization trials and highly similar test trials.

### Familiarization trials

Aggressor, protector, and victim condition. The familiarization videos in the aggressor, protector, and victim conditions consisted of a protective third-party intervention event adapted from past research in which one agent (the aggressor) hits and chases another agent (the victim) before a third agent (the protector) intervenes to protect the victim[29]. The agents were 2-dimensional shapes with eyes. The aggressor was a yellow circle, the victim a blue circle, and the protector a red square. We chose to use geometric shapes as they have been validated in previous studies[29], they are commonly used to investigate socio-moral development[18,36,59], and because they allow greater experimental control of perceptual features between conditions. Furthermore, findings relying on animated agents and puppets and findings relying on human actors tend to converge[60], including studies examining infants' understanding of agency, morality, and moral character[10,14,21,32,59,61].

Each video was 32 s long and began with the aggressor in the top left of the screen, the victim in the top center of the screen, and the protector in the center of the screen surrounded by a white octagonal cage. For the duration of the video, the aggressor chased the victim in a clockwise direction around the cage two times while the protector moved in conjunction with the other two agents as if "following" them. Each time the aggressor contacted the victim, or when the aggressor squished the victim against a wall, one of two sound effects would play. While this was happening, the cage surrounding the protector partially opened at 9 s into the video, and then fully opened at 31 s into the video. The cage opening was accompanied by a different sound effect.

In the final 2 s of the video, the cage fully opened, and the aggressor began accelerating towards the victim one more time. However, before the aggressor could hit the victim, the protector moved out of the cage between the aggressor and the victim. The video stopped once the aggressor contacted the protector, and a ding sound played to indicate that the looking time coding should begin.

Bystander condition. In the bystander condition, the familiarization videos are identical to aggressor, victim, and protector conditions except that the protector is replaced with the bystander. Instead of following the aggressor

and the victim, the bystander moves to "avoid" the altercation by staying on the opposite side of the cage to the aggressor and victim. At the end of the video, the bystander leaves through the bottom of the cage, the opposite side as the aggressor and victim. The video ends after the aggressor hits the victim a final time.

Random movement condition. The familiarization videos in the random movement condition used the same three agents from the aggressor condition and the cage in the center opened in the same pattern. Each of the agents moved in random directions, but they accelerated and decelerated in a similar manner to the agents in the aggressor condition. The agents do not make contact. The yellow agent ends in between the top left and top center of the screen, the blue agent in between the top right and top center of the screen, and the red agent ends slightly above the center of the screen.

### Test trials

Aggressor, protector, victim, and bystander conditions. In these test trials, depending on the condition, the aggressor, protector, victim, or bystander from the familiarization trials distributed four strawberries between two novel recipients, green triangles. Each video began with the distributor at the top of the screen above four strawberries. The two recipients were an equal distance away from the distributor to the bottom right and left, and below the strawberries. In the equal test trial, the distributor first distributed two strawberries to the agent on the left, then two strawberries to the agent on the right. In the unequal test trial, the distributor distributed three strawberries to the agent on the left, then one strawberry to the agent on the right. The order of the test trials was counterbalanced.

Random movement condition. In these test trials, one of the agents from the familiarization trials (either the yellow circle, red square, or blue circle) performed the two resource distributions. The order of the test trials was counterbalanced, and the agent performing the resource distributions was randomized.

**Coding and reliability**. Looking time was live-coded by a human coder using the program jHab[62]. Two trained coders, who were unaware of the condition and order of each test event, recoded a random set of 80 (50%) of the experimental sessions. Intraclass Correlation (ICC) for two random raters, and absolute agreement was calculated to measure reliability. An ICC of 0.98 (95% CI [0.97, 0.98]) was achieved between coder one and the original experimenter and an ICC of 0.93 (95% CI [0.92, 0.94]) was achieved between coder two and the original experimenter. Forty-seven videos overlapped between coder one and coder two. An ICC of 0.96 (95% CI [0.95, 0.97]) was achieved between coder one and coder two.

Further analyses are included in the Supplemental Materials to confirm that infants' looking time in the familiarization trials did not significantly affect their looking time in the test trials (Supplementary Methods 2).

### Experiment 2

**Participants**. We recruited 96 infants aged 12- to 24-months ($M_{age}$ = 17 months, 17 days, $SD$ = 3 months, 16 days, Female = 47, Male = 49; parent-reported) from a participant database at a large North American University and from the Children Helping Science Platform. We chose this sample size based on an a priori power analysis conducted using G*Power 3.1.9.7. We aimed to achieve a power of 0.8 based on an effect size of $d$ = 0.43, the effect size of the difference between fair and unfair looking in the aggressor condition of experiment 1. Children were randomly assigned to either the aggressor or the protector condition. Parents were also sent a social network survey, which was filled out by 44 out of 48 parents in the protector condition, and 42 out of 48 parents in the aggressor condition.

Caregivers reported that infants were white ($n$ = 24), East Asian ($n$ = 23), Mixed ($n$ = 21), South Asian ($n$ = 8), Latin, Central or South American ($n$ = 7), Southeast Asian ($n$ = 3), and Black ($n$ = 2), or did not report ($n$ = 8). Twenty-two additional infants participated but were

excluded from data analysis because of parental interference ($n$ = 2), other environmental interference ($n$ = 2), technical error ($n$ = 2), experimenter error ($n$ = 3), or because infants were inattentive or fussy ($n$ = 13).

**Procedure and stimuli**. This study was a direct replication of the aggressor and protector conditions in Experiment 1. The procedure and stimuli closely mirrored Experiment 1, with infants seeing the same familiarization trials and test trials. The only addition to Experiment 2 was that parents also filled out a social experience survey. This survey measured infants' daycare attendance, number of siblings, number of caregivers, and frequency of face-to-face interactions.

**Coding and reliability**. Looking time was live-coded by a human coder using the program jHab[62]. Two trained coders, who were unaware of the condition and order of each test event, recoded all the experimental sessions except for four. Three participants did not consent to recording, and one video was lost due to a technical error. Intraclass Correlation (ICC) for two random raters, and absolute agreement was calculated to measure reliability. An ICC of 0.97 (95% CI [0.96, 0.97]) was achieved between coder one and the original experimenter and an ICC of 0.95 (95% CI [0.95, 0.96]) was achieved between coder two and the original experimenter. An ICC of 0.95 (95% CI [0.94, 0.95]) was achieved between coder one and coder two.

## Results
### Experiment 1

Our general analytic approach was preregistered at https://aspredicted.org/CN8_5SD. All tests reported are two-sided.

To test our main hypothesis concerning whether infants can make moral character inferences in protective third-party intervention events, we conducted a mixed ANOVA specifying condition (aggressor vs. protector vs. bystander vs. victim vs. random movement) as a between-subjects factor, test trial type (fair vs. unfair) as a within-subjects factor and looking time as the dependent variable. Consistent with the moral character inference hypothesis, we found that there was a significant interaction effect between condition and test trial type ($F(4, 155)$ = 5.47, $p < 0.001$, $\eta^2_p$ = 0.12, 95% CI [0.03, 0.21]), suggesting that infants' fairness expectations for each agent differed based on the previous behavior of that agent. There was not a significant main effect of condition ($F(4, 155)$ = 1.90, $p$ = 0.11, $\eta^2_p$ = 0.05, 95% CI [0.00, 0.11]), nor was there a significant main effect of test trial type ($F(1, 155)$ = 3.82, $p$ = 0.052, $\eta^2_p$ = 0.02, 95% CI [0.00, 0.09]).

Next, we aimed to examine infants' fairness expectations towards each of the agents in the protective third-party intervention event. For the following analyses, paired t-tests were conducted in each condition to compare infants' looking time towards the fair and unfair test trials. Looking times toward each test trial are displayed in Fig. 2. Binomial tests were also conducted to determine if there were differences in the number of infants looking longer toward the unfair over the fair test trial, a measure commonly reported in violation-of-expectation studies[14]. A participant is deemed to have looked longer at the unfair test trial if their looking time towards the unfair test trial was at least 1 s greater than their looking time towards the fair test trial, and vice versa. Infants whose looking to the fair and unfair test trials did not differ by more than 1 s were excluded from this analysis.

In the aggressor condition, infants looked significantly longer at the fair ($M$ = 13.37, $SE$ = 1.30) test trial than the unfair ($M$ = 9.26, $SE$ = 1.24) test trial ($t(31)$ = 2.45, $p$ = 0.020, $d$ = 0.43, 95% CI [0.07, 0.79]). Additionally, 7 infants looked longer at the unfair test trial and 24 infants looked longer than the fair test trial. An exact binomial test demonstrates that this proportion significantly differs from chance ($p$ = 0.003). In the protector condition, infants looked significantly longer at the unfair ($M$ = 16.22, $SE$ = 1.59) test trial than the fair ($M$ = 10.71, $SE$ = 1.22) test trial ($t(31)$ = 3.58, $p$ = 0.001, $d$ = 0.63, 95% CI [0.25, 1.01]). Twenty-two infants looked longer at the unfair test trial and 8 infants looked longer at the fair test trial. An exact binomial test demonstrates that this proportion significantly differs from chance ($p$ = 0.016). Consistent with our hypotheses, these findings

demonstrate that infants expected agents that previously behaved aggressively (by chasing and hitting another agent) to behave unfairly, but they expected an agent that previously protected another agent (by coming in between an aggressor and a victim) to distribute resources fairly.

Next, we examined infants' expectations towards two morally ambiguous agents: victims and bystanders. In the bystander condition, there was no significant difference in looking time towards fair ($M = 9.91$, SE = 1.24) test trials and unfair ($M = 8.62$, SE = 1.39) test trials ($t(31) = 0.78$, $p = 0.44$, $d = 0.14$, 95% CI [$-0.21$, 0.49]). Nine infants looked longer at the unfair test trial, and 16 looked longer at the fair test trial. An exact binomial test demonstrates that this proportion does not significantly differ from chance ($p = 0.22$). In the victim condition, infants looked significantly longer at the unfair ($M = 13.53$, SE = 1.59) test trial than the fair ($M = 9.53$, SE = 1.37) test trial ($t(31) = 2.12$, $p = 0.042$, $d = 0.37$, 95% CI [0.01, 0.73]). Eighteen infants looked longer at the unfair test trial, and 7 looked longer at the fair test. An exact binomial test demonstrates that this proportion significantly differs from chance ($p = 0.04$). These findings suggest that there were no significant differences in infants' reactions when bystanders behaved fairly vs. unfairly, and that infants expected victims to behave fairly.

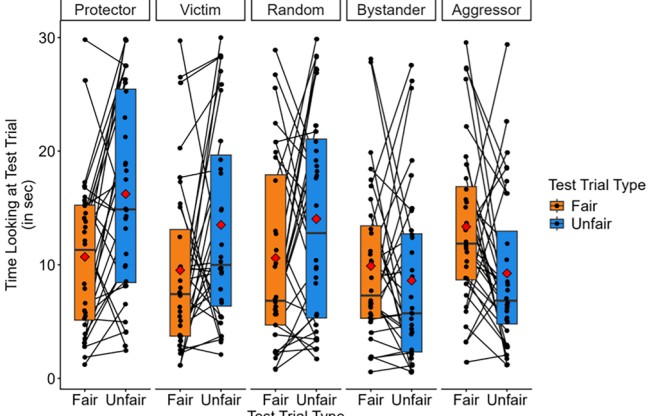

**Fig. 2 | Experiment 1 looking time by condition and test trial.** $n = 32$ participants per condition. Points and lines representing looking time and difference in looking time for each participant are overlaid over box plots (1st quartile, median, 3rd quartile). The red diamonds represent condition means. Paired sample t-tests demonstrated significant differences in looking time in the aggressor, protector, and victim conditions.

Infants also participated in a random movement condition. Instead of being familiarized to a protective third-party intervention event as in the other conditions, in this condition, infants were familiarized to three agents moving in random directions. The agents' shape, color, and manner of movement were identical to the other conditions. Following this, infants observed one of the three agents engaging in fair and unfair resource distributions. Infants looked marginally, but not significantly longer towards the unfair ($M = 14.04$, SE = 1.66) test trial over the fair ($M = 10.61$, SE = 1.47) test trial ($t(31) = 1.84$, $p = 0.076$, $d = 0.33$, 95% CI [0.03, 0.68]). Seventeen infants looked longer at the unfair test trial and 12 infants looked longer at the fair test trial. An exact binomial test demonstrates that this proportion does not significantly differ from chance ($p = 0.46$). Additionally, an ANOVA demonstrated that infants' looking time did not significantly change based on the color and shape of the agent ($F(2, 29) = 0.63$, $p = 0.54$, $\eta^2_p = 0.04$).

**Comparisons between conditions.** Based on our preregistered analyses, we determined that infants' fairness expectations differed based on the previous agents' role in a protective third-party intervention event and infants' fairness expectations for agents in each of these roles. The goal of the following exploratory analyses is to compare between each of the conditions to determine which roles infants were able to differentiate between. To do this, we calculated a proportion score for each participant from the proportion of time infants spent looking at the unfair distribution compared to their total looking time at both test trials (proportion score = unfair looking time/(fair looking time + unfair looking time)). In this case, higher proportion scores demonstrate that infants were relatively more surprised when the agent behaved unfairly than fairly. Following this, we conducted pairwise comparisons using two-sample t-tests to investigate which agents infants were able to distinguish between. A summary of these comparisons, in addition to p-values adjusted for multiple comparisons using the Tukey HSD method, can be found in Table 1. A proportion score was chosen as it controls for overall looking time differences between participants and is a commonly used measure in looking time studies[53]. Using a difference score (fair–unfair looking time) yields similar results, with one notable difference being that the comparison between the bystander and random movement condition becomes marginally significant instead of significant (Supplementary Note 1).

As Table 1 and Fig. 3 indicate, infants' fairness expectations appeared to be ordered descriptively along a moral continuum, with statistically significant differentiation between protectors versus bystanders and aggressors, victims versus bystanders and aggressors, and random agents versus

**Table 1 | Experiment 1 exploratory comparisons between conditions**

| Agent | Protector | Victim | Random | Bystander | Aggressor |
|---|---|---|---|---|---|
| Protector | - | $d = 0.10$<br>95%CI<br>[$-0.39$, 0.59] | $d = 0.15$<br>95%CI<br>[$-0.34$, 0.65] | **$d = 0.82$**<br>**95%CI**<br>**[0.31, 1.33]** | **$d = 1.10$**<br>**95%CI**<br>**[0.57, 1.63]** |
| Victim | $t = 0.41$<br>$p = 0.68$<br>$p_{adj.} = 0.99$ | - | $d = 0.04$<br>95%CI<br>[$-0.44$, 0.54] | **$d = 0.65$**<br>**95%CI**<br>**[0.14, 1.15]** | **$d = 0.88$**<br>**95%CI**<br>**[0.37, 1.39]** |
| Random | $t = 0.62$<br>$p = 0.54$<br>$p_{adj.} = 0.98$ | $t = 0.18$<br>$p = 0.86$<br>$p_{adj.} = 0.99$ | - | **$d = 0.62$**<br>**95%CI**<br>**[0.11, 1.12]** | **$d = 0.86$**<br>**95%CI**<br>**[0.34, 1.37]** |
| Bystander | **$t = 3.30$**<br>**$p = 0.001$**<br>**$p_{adj.} = 0.015$** | **$t = 2.59$**<br>**$p = 0.012$**<br>**$p_{adj.} = 0.048$** | **$t = 2.47$**<br>**$p = 0.016$**<br>$p_{adj.} = 0.078$ | - | $d = 0.21$<br>95%CI<br>[$-0.28$, 0.70] |
| Aggressor | **$t = 4.41$**<br>**$p < 0.001$**<br>**$p_{adj.} < 0.001$** | **$t = 3.53$**<br>**$p < 0.001$**<br>**$p_{adj.} = 0.003$** | **$t = 3.43$**<br>**$p = 0.001$**<br>**$p_{adj.} = 0.006$** | $t = 0.83$<br>$p = 0.41$<br>$p_{adj.} = 0.91$ | - |

Paired sample t-tests were conducted to compare the proportion scores in each condition. Bolded values are used to indicate statistically significant comparisons. The values on the bottom left show t-values ($df = 62$ for all comparisons), raw p-values, and Tukey HSD adjusted p-values. The values on the top right show effect sizes and 95% confidence intervals.

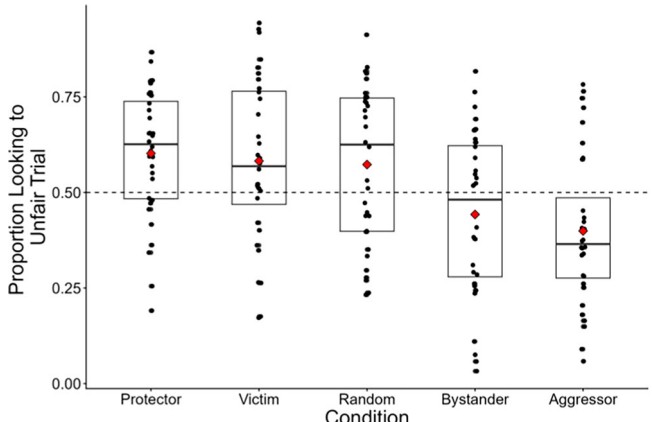

**Fig. 3 | Proportions scores by condition.** $n$ = 32 participants per condition. Higher proportion scores represent greater looking to the unfair test trial. Points representing looking time for each participant are overlaid over box plots (1st quartile, median, 3rd quartile). The red diamonds represent condition means, and the dashed line represents equal looking.

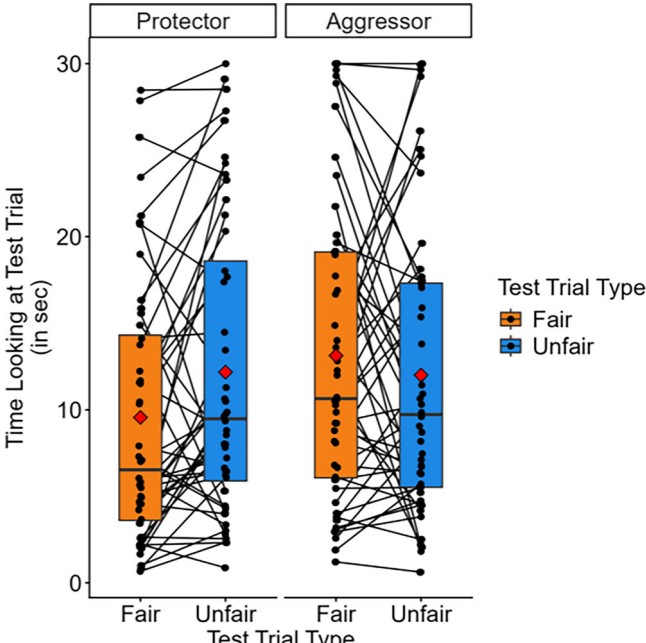

**Fig. 4 | Experiment 2 looking time by condition and test trial.** $n$ = 48 participants per condition. Points and lines representing looking time and difference in looking time for each participant are overlaid over box plots (1st quartile, median, 3rd quartile). The red diamonds represent condition means. Paired sample t-tests demonstrated significant differences in looking time in the protector condition.

bystanders and aggressors. An alternate method for analyzing our data is presented in Supplementary Note 2.

## Experiment 2

In Experiment 2, we aimed to replicate two key conditions from Experiment 1 (the aggressor and protector conditions) and investigate the effect of individual differences, such as age and daycare/sibling status, on infants' moral character inferences. Our analytic approach for Experiment 2 was preregistered here: https://aspredicted.org/yi9u4y.pdf.

Mirroring the main analyses of Experiment 1, we conducted a mixed ANOVA specifying looking time as the dependent variable, condition (aggressor vs. protector) as a between-subjects factor, and test trial type (fair vs. unfair) as a within-subjects factor (Fig. 4). This analysis replicated the

results of Experiment 1, finding a significant interaction between condition and test trial type ($F(1, 94) = 5.51$, $p = 0.021$, $\eta^2_p = 0.06$, 95% CI [0.00, 0.17]), and no significant main effect of condition ($F(1, 94) = 1.27$, $p = 0.26$, $\eta^2_p = 0.01$, 95% CI [0.00, 0.09]) or test trial type ($F(1, 94) = 0.89$, $p = 0.35$, $\eta^2_p = 0.00$, 95% CI [0.00, 0.08]).

Infants in the protector condition looked significantly longer towards the unfair ($M = 11.78$, SE = 1.20) test trial than the fair ($M = 9.13$, SE = 1.08) test trial ($t(47) = 2.65$, $p = 0.01$, $d = 0.38$, 95% CI [0.09, 0.67]), replicating the results of Experiment 1. Contrary to Experiment 1, infants' looking in the aggressor condition did not significantly differ between the unfair ($M = 12.02$, SE = 1.22) test trial and the fair ($M = 13.14$, SE = 1.25) test trial ($t(47) = 0.90$, $p = 0.37$, $d = -0.13$, 95% CI [−0.41, 0.16]), though looking did numerically trend in the same direction as Experiment 1.

Binomial tests revealed that in the protector condition, 28 infants looked longer than 1 s toward the unfair test trial, and 13 infants looked longer towards the fair test trial ($p = 0.028$), but in the aggressor condition, 22 infants looked longer towards the unfair test trial and 19 looked longer towards the fair test trial ($p = 0.76$). An additional t-test demonstrated that the proportion of time that infants spent looking at the fair test trial significantly differed between the aggressor and protector conditions ($t(94) = 2.64$, $p = 0.01$, $d = 0.54$, 95%CI [0.13, 0.95]), mirroring the exploratory analysis in Experiment 1.

**Daycare and sibling analyses.** In addition to our main analyses of interest, we also conducted exploratory analyses examining the effects of sibling and daycare experience on infants' looking patterns. To do this, we began by creating a composite score of infants' daycare experience and sibling information. We chose these two aspects of infants' early social environment because they were likely to reflect the number of peer relationships that infants had, a factor that is particularly important to infants' cognitive development[51,52], and because daycare and sibling information could be accurately measured on the short self-report scale that we administered to parents.

Infants with no daycare experience were coded as 0, part-time daycare experience were coded as 0.5, and full-time daycare experience were coded as 1. Infants with no siblings were coded as 0 and infants with siblings were coded as 1. The daycare/sibling composite score was calculated by taking the sum of the infant's daycare experience score and sibling score. Additionally, we also calculated a proportion score from the proportion of time that infants looked at the unfair vs. the fair test trial. This was done to control for infants' total looking time and is consistent with past research examining the effect of individual differences on looking time measures[53].

Next, we ran a linear model specifying condition (protector vs. aggressor), daycare/sibling score, and their interaction as predictors, and proportion score as the dependent variable. This analysis found a significant interaction effect between condition and daycare/sibling score ($\beta = 0.11$, 95% CI [0.01, 0.22], se = 0.05, $t(82) = 2.27$, $p = 0.026$; Fig. 5), such that infants with more social experience demonstrated greater differentiation by condition. There were no significant main effects of condition or daycare/sibling score. Follow-up analyses revealed that infants that had daycare experience and siblings (i.e., the maximum of the composite score) differentiated between conditions ($t(15) = 2.24$, $p = 0.04$, $d = 1.09$, 95% CI [0.05, 2.10]), but infants with no daycare experience or siblings (i.e., the minimum of the composite score) did not significantly differentiate between conditions ($t(15) = 0.65$, $p = 0.52$, $d = 0.24$, 95% CI [−0.49, 0.98]).

We also conducted an additional linear regression treating siblings and daycare experience as two separate variables, including the interaction between condition and siblings, and the interaction between condition and daycare experience. Although infants' looking trended in the same direction as in the composite score for both of these variables, their looking was not significantly predicted by the interaction between condition and sibling presence ($\beta = 0.13$, 95% CI [−0.03, 0.29], se = 0.08, $t(80) = 1.66$, $p = .10$), nor by the interaction between condition and daycare experience ($\beta = 0.10$, 95% CI [−0.06, 0.26], se = 0.08, $t(80) = 1.28$, $p = 0.20$).

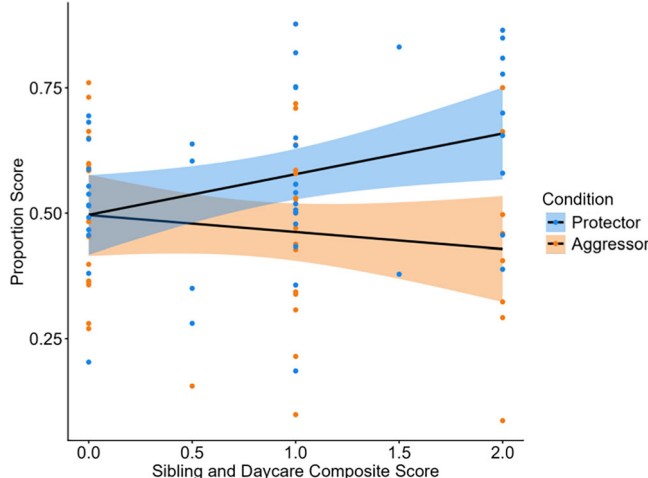

**Fig. 5 | Proportions scores by sibling/daycare experience and condition.** $n = 44$ participants in the protector condition and $n = 42$ participants in the aggressor condition. Higher proportion scores represent greater looking at the unfair test trial. Proportion scores are represented by points, and the shaded area represents 95% confidence intervals.

Next, we sought to investigate the effect of sibling and daycare experience in each condition. In the protector condition, infants' looking was significantly predicted by their daycare/sibling score ($\beta = 0.08$, 95% CI [0.01, 0.15], se = 0.03, $t(42) = 2.36$, $p = 0.023$). In contrast, infants' looking in the aggressor condition was not significantly predicted by daycare/sibling score ($\beta = -0.03$, 95% CI [−0.11, 0.04], se = 0.04, $t(40) = 0.91$, $p = 0.37$).

**Age analyses**. Given that our study had a large age range, we also conducted exploratory analyses examining the effect of age on infants' character inferences. To do this, we combined the data from Experiments 1 and 2 and ran linear mixed effects models examining the effect of age on proportion scores, adding Experiment (1 vs. 2) as a random effect. In the protector condition, we found that there was no significant effect of age on infants' looking ($\beta = -0.004$, 95% CI [−0.015, 0.006], se = 0.005, $t(78) = 0.81$, $p = 0.42$). In the aggressor condition, there was also no significant effect of age on infants' looking ($\beta = -0.01$, 95% CI [−0.023, 0.002], se = 0.006, $t(78) = 1.58$, $p = 0.12$). Similar analyses conducted on the other three conditions in Experiment 1 found no significant ages effects (victim: $\beta = -0.01$, 95% CI [−0.03, 0.01], se = 0.01, $t(30) = 1.17$, $p = 0.25$, bystander: $\beta = -0.003$, 95% CI [−0.024, 0.017], se = 0.01, $t(30) = 0.34$, $p = 0.74$, random: $\beta = 0.007$, 95% CI [−0.016, 0.031], se = 0.01, $t(30) = 0.61$, $p = 0.55$).

**Daycare, siblings, and age**. Given that infant's age and sibling/daycare experience are highly correlated due to increasing daycare attendance with age ($\beta = 007$, 95% CI [0.03, 0.12], se = 0.02, $t(82) = 3.37$, $p = 0.001$), we next sought to parcel out the independent effect of each variable. To test this question, we ran a multivariate linear regression including condition, age, daycare/sibling score, the interaction between daycare/sibling score and condition, and the interaction between age and condition. This analysis demonstrated that the interaction between condition and daycare/sibling score remained significant ($\beta = 0.15$, 95% CI [0.05, 0.25], se = 0.05, $t(80) = 2.85$, $p = .005$), but that there was no significant interaction between condition and age ($\beta = -0.02$, 95% CI [−0.04, 0.01], se = 0.01, $t(80) = 1.46$, $p = 0.15$).

## Discussion

Consistent with a moral character inference hypothesis, infants' fairness expectations for an agent differed based on that agent's previous behavior across both of our Experiments. Infants expected protectors to be fair, but held different expectations for aggressors, either actively expecting

aggressors to subsequently be unfair (Experiment 1) or displaying no significant differences in their looking (Experiment 2). Our results also provide information about infants' moral character inferences for agents that act in morally ambiguous ways. Infants expected victims to distribute resources fairly but did not significantly differ in their looking towards bystanders, and they were also able to distinguish protectors and victims from bystanders and aggressors.

In comparison to prior work suggesting that infants have baseline expectations for fairness[21], when infants see agents engage in random movement before distributing resources, their expectations of subsequent fair behavior are attenuated: although infants in the random movement condition trended toward longer looking to unfair versus fair outcomes, no significant differences in infants' looking times between the test events were obtained. This may have been because randomly moving agents violate certain features associated with agency, such as rational and goal directed action[14,61], and thus, infants expected these agents to act less typically than agents that infants had no previous information about.

Although infants' positive character inferences towards protectors and negative character inferences towards aggressors align with lay intuitions about these roles, infants' character inferences towards victims and bystanders were more surprising. Adults have mixed intuitions towards victims, sometimes viewing them as virtuous victims, but also sometimes engaging in victim blaming[35,42]. Our findings suggest that infants, at least in the current context, treat victims as morally good. Evaluating the moral inaction of a bystander is much more challenging than evaluating moral actions[45]. Despite this, infants may be able to make moral character inferences regarding bystanders, as their looking did not significantly differ when bystanders behaved fairly or unfairly, but it did when victims or protectors performed the same distributions. It is not clear, however, whether these results are due to the moral inaction of the bystander or if they are the result of the bystander's "cowardly" nature and avoidance of the aggressor and victim. Nevertheless, these results provide insights into the developmental origins of infants' capability to evaluate moral inaction.

Taken together, our findings demonstrate that infants can make moral character inferences in multi-agent scenarios containing morally complex agents rather than just dyadic scenarios containing prototypically good or bad agents, as has been standard in previous research[10,31,32]. Infants' capacity to make inferences concerning morally ambiguous agents also supports the notion that infants may view others' character along a moral continuum rather than making categorical good vs bad judgments[31].

Our investigation into infants' character inferences provides insight into ongoing debates about the primacy of moral actions versus moral character in adults' moral judgments[63]. This body of work has demonstrated that judgments of moral character and moral actions are dissociable[64,65], judgments of moral character are core to person perception[7], and judgments of moral character may better explain phenomena such as moral dumbfounding[66]. This has led researchers to conclude that evaluating others' moral character may be more important to humans than judging others' moral actions. Our findings are consistent with this general notion as we document the emergence of moral character inferences coincident with, or only slightly after, infants' ability to understand and evaluate non-moral and moral actions[14,18].

We also demonstrate that the ability to make character inferences is an early emerging capacity, present within the second year of life. This was particularly surprising as even young children can struggle to generalize behavior following a single example[3]. These discrepancies may be explained via differences in task demands: studies conducted with children use verbal responses, while our study used an implicit measure (i.e., looking time). Consistent with this claim, if children's generalization is measured via surprise to consistent versus inconsistent moral behaviors, even young children show evidence of moral generalization after seeing a single exemplar[67].

Another unique contribution of our work was that our larger sample size and age range allowed us to test the effect of individual difference

measures such as sibling/daycare experience and age on infants' moral character inferences, questions which previous research had not investigated[10,31,32]. In Experiment 2, we found that although daycare and sibling experience did not independently predict infants' looking, the cumulative effect of these two variables was related to a higher propensity for making moral character inferences. Infants with siblings and daycare experience were more likely to have different fairness expectations between conditions, suggesting that infants' early social environment may play a role in the development of infants' moral character inferences. Siblings and daycare experience might enhance infants' moral role differentiation in two ways. First, to the extent that infants' peers tend to behave in morally consistent ways across contexts, social contact with peers may help infants construct stable moral character representations of other agents. Second, making moral character inferences allows infants to represent other agents along a single good-bad moral continuum rather than multiple specific moral trait dimensions, thereby saving cognitive resources. The benefits of this reduction in cognitive resources may be felt more strongly by infants with siblings and in daycare because they have more social contacts to represent[68].

We also tested for age effects in our study. Across both experiments, infants' moral character inferences were generally consistent across the age range tested (12 to 24 months), and age was not a significant predictor of infants' looking in any of the conditions. However, given that sibling/daycare experience and age are highly correlated, as older infants are also more likely to be in daycare, we next examined the effect of these two variables in tandem. Siblings/daycare experience continued to be a significant predictor of condition differentiation, but there were no significant effects of age. This raises the possibility that social experience rather than age-based neural maturation is a larger driver of infants' moral character inferences, supporting experience-dependent theories of socio-moral development[27].

These individual difference-related changes may also explain the discrepancy that we found between aggressor condition in Experiment 1, where infants expected aggressors to be unfair, and Experiment 2, where infants did not hold expectations for aggressors. During the transition between stages of understanding, cognitive abilities can often enter periods of high instability, becoming more susceptible to sampling variation and decreasing replicability[69,70]. It is possible that the smaller sample size in Experiment 1 led to increased sampling variance. This could have resulted in differences in infants' social experience between the two studies, which also may have driven the differences in infants' looking.

## Limitations

Some of the characteristics of our sample may have limited the interpretability of our findings. Participants in our study were mainly based in a large North American urban center. Research has demonstrated, however, that there is significant cross-cultural variability in infants' moral development. For example, infants possess baseline fairness expectations in a variety of Western cultures, but research demonstrates that this may not be the case for the Samburu and Kikuyu populations in Kenya[71]. Without additional data investigating infants' abilities to make moral character inferences in different cultures and social contexts, our understanding of moral character inferences remains incomplete.

Although we have demonstrated that infants can make moral character inferences, future research should attempt to disentangle the scope of these inferences. Do infants also make character inferences in non-moral domains, or is the moral domain "privileged" as some have suggested[7,31]. What is the scope of infants' character inferences? Do infants see an immoral agent as generally "negative", and thus also expect negative behavior in non-moral domains (e.g., incompetence)? Or do infants make more specific inferences, inferring that an immoral agent is only "immoral", and restricting their generalizations to other moral transgressions? Future work can address these critical questions, which will enrich our understanding of infants' moral character inferences.

## Conclusions

In sum, we provide evidence that, after seeing multi-agent social interactions, infants subsequently form moral expectations consistent with the agents' prior moral roles, suggesting that infants make moral character inferences much like older children and adults would. These character inferences appear to be supported by the infant's social experience. Future work should investigate converging measures of infants' capabilities and further address the role of individual differences and cultural context in infants' character inferences.

## Data availability
Our data can be found at the following link: https://osf.io/gjd3p/overview

## Code availability
The code for our analysis can be found at the following link: https://osf.io/gjd3p/overview

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

## Acknowledgements
This work was funded by the Social Sciences and Humanities Research Council of Canada (SSHRC Insight Grant 435–2022-0118). The funders had no role in study design, data collection and analysis, decision to publish or preparation of the manuscript. We would like to thank Samarrann Sivaloganathan, Lily Huang, Aaron Wang, Vera Zhang, and Natalie Cheung for their assistance in data collection and reliability coding.

## Author contributions
Norman J. Zeng contributed to conceptualization, methodology, formal analysis, investigation, writing—original draft, writing—reviewing and editing, visualization, and project administration. Inderpreet K. Gill contributed to conceptualization, methodology, and writing—reviewing and editing. Jessica A. Sommerville contributed to conceptualization, methodology, resources, writing—reviewing and editing, supervision, and funding acquisition.

## Competing interests
The authors declare no competing interests.
