## [Transparent Peer Review File · Communications Psychology]

Infants Make Moral Character Inferences in Multi-Agent Social Interactions

Corresponding Author: Mr Norman Zeng

Version 0:

Decision Letter:

Dear Mr Zeng,

Thank you for your patience during the peer-review process. Your manuscript titled "Infants Make Moral Trait Inferences in Multi-Agent Social Interactions" has now been seen by 3 reviewers, and I include their comments at the end of this message. They find your work of interest but raised some important points. We are interested in the possibility of publishing your study in Communications Psychology, but would like to consider your responses to these concerns and assess a revised manuscript before we make a final decision on publication.

We therefore invite you to revise and resubmit your manuscript, along with a point-by-point response to the reviewers. Please highlight all changes in the manuscript text file.

Editorially, we consider it important that the revised manuscript address the reviewers' concerns regarding the inference of specific moral traits by the participants and better situate the findings within the existing literature. Please note that while you may make revisions to clarify the terminology, we are not asking for a reframing of the study. We encourage you to conduct the additional analyses as suggested by Reviewer 3, however as the study was preregistered, please also report the originally planned analyses (for our policy on preregistration please see: <https://www.nature.com/commspsychol/editorial-policies/preregistration-policy>). Preregistered analyses should not be relegated to the Supplementary Information.

Please ensure you follow our statistical guidelines when reporting statistics (<https://www.nature.com/commspsychol/submit/submission-guidelines#statistical-guidelines>). Please note in particular our requirements for the reporting and interpretation of null-results. Non-significant findings derived from null-hypotheses significance tests should be reported in full, but may not be interpreted. Where you interpret null results, this interpretation must be based on Bayes Factors or equivalence tests.

I am attaching an Editorial Requests Table that details critical reporting requirements for the revised manuscript. Please attend to each item and ensure your manuscript is fully compliant. If your revised manuscript is not aligned with these requests on major issues, such as those concerning statistics, it may be returned to you for further revisions without re-review.

Please submit the following items:

- Revised manuscript
- Point-by-point response to the referees' comments
- Cover letter (as a separate document)

- <https://www.nature.com/documents/nr-reporting-summary.zip>>Nature Research Reporting Summary
- <https://www.nature.com/documents/nr-editorial-policy-checklist.pdf>>Editorial Policy Checklist
- Completed Editorial Request Table (attached).

via this link: Link Redacted .

Additional guidance is available in our style and formatting guide <https://www.nature.com/documents/commspsychol-style-formatting-guide-accept.pdf>>Communications Psychology formatting guide.

Best regards,

Jennifer Bellingtier

Jennifer Bellingtier, PhD
Senior Editor
Communications Psychology

REVIEWER EXPERTISE:

Reviewer #1 social cognition, developmental psychology
Reviewer #2 social cognition, developmental psychology
Reviewer #3 social cognition, developmental psychology

REVIEWER REPORTS:

Reviewer #1 (Remarks to the Author):

In this study, the authors attempted to demonstrate infants' inferences of moral traits in multi-agent social situations in 5 conditions. They found that infants had different expectations depending on the previous role. This finding is very interesting to me and would be important in this area of research. However, I am not convinced that the authors can demonstrate trait inference from the results of the experiments, especially the negative one (aggressor condition). In the aggressor condition, the prior role of the agents is aggressor, which is attributed to physical traits such as strong or violent, rather than the moral trait. So I am not sure about the link between the aggressor and the unfair distributor. How can the authors explain the link between them in terms of the moral trait?

Related to this point, if the authors can explain the link, they could discuss why they found the negative trait inference, which has not been found in previous studies (Surian et al., 2018; Gill & Sommerville, 2023).

Another concern is the wide age range of the participants. The authors should justify in the manuscript why they chose such a wide range of ages.

Minor points;

- Line 82, the authors misunderstand the findings of Kanakogi et al. (2017).
- Line 98-101, I am not sure about the first limitation. Please add the example.
- The authors have overlooked the study by Kanakogi et al. (2013) in their reference.
- Line 375-377, why did the authors analyse the violation of expectations in the familiarisation phase?

Reviewer #2 (Remarks to the Author):

This study presents a thorough examination of infants' abilities to infer moral trait in complex social scenarios. The researchers have employed looking time methodology to study how infants aged 12 to 24 months form expectations about fairness based on agents' prior roles in multi-agent interactions. The experimental design is thorough including conditions for aggressors, protectors, victims, bystanders, and a random movement control that allows for detailed comparisons between different social roles. The statistical analyses presented in this manuscript are appropriate and well-executed. The lack of age effects across the age range suggests that the ability to make moral trait inferences may emerge earlier than previously thought. The results indicate that character inferences may be foundational in infancy.

However, there are some aspects to consider. The use of abstract geometric shapes as agents, while allowing for precise control of stimuli, could constrain the validity. Most, but not all previous studies with infants and toddlers have used human actors in test stimuli. The authors should elaborate on how their approach using geometric shapes differs from these prior methods and discuss potential implications for interpreting the results. Additionally, the reliance on looking time as the only dependent measure, while well-established in infant research, has limitations. The authors should more thoroughly discuss the validity of using looking time to infer infants' moral expectations and consider potential alternative explanations for the observed patterns. Furthermore, the authors could situate their findings in relation to previous studies on infants' and toddlers' fairness reasoning, examining how the lack of age effects aligns with or differs from prior work. Finally, the discussion section would benefit from an expanded consideration of the universality versus cultural specificity of these moral trait inferences. Given that moral values and judgments can vary significantly across societies, it is crucial to address whether the observed abilities in infants from a Western cultural context would generalize to infants from different cultural backgrounds.

In sum, this study makes a valuable contribution to our understanding of early moral cognition. The findings are novel and have significant implications for developmental psychology. With some revisions to address the limitations noted, this research advances the field of the developmental origins of moral reasoning.

Reviewer #3 (Remarks to the Author):

With some revision, this manuscript should make a very nice contribution to the developmental literature. The main findings are interesting and thought-provoking, though some re-writing is needed to make this clear.

Main points

1. One issue has to do with the framing used in the Abstract and Introduction, which does not serve the research well. First, it is not clear to me that the present research has to do with infants' ability to infer specific moral traits (e.g., honesty, kindness, humility, courage, and so on); a broader focus on moral character would, I think, serve the research better. I may be wrong, but I suspect that many readers would view an agent's moral character less as a specific moral trait and more as a global assessment of the agent's moral disposition.

Second, the authors need to be more straightforward in presenting prior reports, both to acknowledge their contributions and to make clear what is novel about the present research. Going chronologically, Surian et al. (2018) found that 15-month-olds who first saw A help B achieve a goal (in 2 familiarization trials) and saw C hinder B from achieving this goal (in 2 familiarization trials) were later surprised if A divided resources unfairly between two novel targets (2:0), but looked equally whether C acted fairly or unfairly toward the novel targets. In a similar vein, Ting and Baillargeon (2021) found that after seeing A harm an ingroup's or an outgroup's property in 3 familiarization trials, 25-month-olds looked equally if A next divided resources unfairly between two ingroup targets (2:0), but were surprised if A chose to share resources generously with an ingroup target. Finally, Gill and Sommerville (2023) replicated the findings of Surian et al. with 14- to 27-month-olds, using a 5:1 violation. In addition, they found that when the events shown in familiarization and test were reversed, with A acting fairly and C unfairly, infants looked equally in test whether A chose to help or to hinder B (since there was no indication that A and B belonged to the same group, either action was acceptable), whereas infants looked reliably longer when C chose to help as opposed to hinder B (helping an outgroup is a supererogatory, virtuous action that one would not expect from an agent with a poor moral character).

Together, these results suggest that when infants take an agent's negative action to provide evidence of a poor moral character (see Ting & Baillargeon, 2021, for discussion of some of the factors that affects such decisions), they then use this inference to predict the agent's actions in new contexts. For example, they are not surprised if an agent who gives evidence of a bad moral character later acts unfairly (the default expectation that agents will be of good character and will act fairly is weakened). Conversely, they are surprised if an agent who gives evidence of a bad moral character later acts virtuously (i.e., goes beyond what is obligated by a good character), for example, by sharing resources generously with an ingroup member, or by kindly helping an individual who is not clearly identified as an ingroup member.

In contrast, when agents' actions do not lead infants to question the goodness of their characters, then typical expectations apply (e.g., agents will act fairly, and they may choose to help or hinder outgroup members as they wish—helping is obligatory only for ingroup members).

Sorry for the long review—it helps situate the present work. What is exciting here is that the authors raise two novel questions that go beyond these prior findings. First, what happens when an agent gives evidence of having not simply a bad character, but a vicious character? This is an issue that was raised by Ting and Baillargeon (2021) in their discussion: "What if the wrongdoer first committed a more severe violation, by directing more harmful actions at the victim or by harming a larger number of victims? Would children now expect her to act unfairly in the test trials, and hence would they look

significantly longer if she acted fairly instead?" (p. 9). In the present research, the harm is directed at the victim itself (i.e., at its body), and it is repeated numerous times. In the prior research, infants and toddlers saw only 2 or 3 instances of hindering or harming (destruction of personal property) to start. But here we have an aggressor who does great bodily harm to a victim: Each familiarization trial lasts 30 s and involves 11 or 12 hits plus 3 squishes! Across the 4 familiarization trials, this represents a great deal of bodily harm (over 40 hits plus 12 squishes), leading infants to attribute a very bad (or vicious) moral character to the aggressor.

Second, what happens when an agent does not commit such harm but allows it to happen without ever intervening? Indeed, the "bystander" continually avoids the aggressor by staying as far away from it as possible, and it eventually comes out on the opposite side of the cage. This all conveys a defensive and perhaps cowardly posture that, to my mind at least, makes the term "bystander" less than adequate. This is not just a passive bystander who observes what is going on; this is a character who appears to fear the aggressor and stays away from it, in trial after trial. Indeed, the results suggest that infants attribute a poor character to the cowardly bystander, who is guilty of harm by omission.

The approach I am suggesting echoes some of the themes the authors lay out in the Discussion. Embracing this approach from the Introduction would have two important benefits: It would make clear how the present research builds on and extends prior findings, and it would allow the authors to make richer and more interesting predictions about the likely results in each condition. As things stand now, it is unclear why the authors predict what they do in the aggressor condition (vicious character), the bystander condition (poor character), the protector condition (virtuous character), and the victim condition (good character).

2. I would suggest more standard analyses of the data that are less likely to raise questions about corrections for multiple comparisons. For example, the authors could have two experiments, one focusing on the aggressor (severe bodily harm by commission) and one on the bystander (severe bodily harm by omission). Experiment 1 would include the aggressor condition (fair > unfair) and the protector condition (unfair > fair, the standard result). After showing that the familiarization data are similar in the two conditions (to make sure these are not directly responsible for any test differences), the authors could focus on the test data, where we would expect a significant interaction between the two conditions. Experiment 2 could have the bystander condition (unfair = fair) and victim condition (unfair > fair), with again a significant interaction between the two conditions. This way, in each experiment, one condition would show the standard result (unfair > fair), and one condition would deviate from it, as predicted.

3. I think the random condition is problematic and should be relegated to the SI. It is not clear what it is controlling for, and it did not yield the expected result of longer looking at the unfair as opposed to the fair event. This suggests that infants were puzzled by the random movements of the characters (what are they doing??) in the 4 familiarization trials, leaving them to suspend their expectations in the test trials.

Since three different looking-time patterns are found in the other four conditions, in a sensible and predictable way, the random control adds little to the conclusions. Moving that control condition to the SI would also help limit the number of comparisons performed.

4. For me, the current analyses, as reported, are problematic. First, as noted, there is no control for multiple comparisons within the same overall analysis, something we might expect with so many comparisons. Second, the analyses of the proportion data are non-standard and are potentially unwarranted (they could be deleted). For example, higher levels of looking might be expected overall in the protector condition, which depicts a virtuous, heroic character, than in the bystander condition, which depicts a cowardly character. The simpler and more standard analyses I suggested above would make these proportion analyses unnecessary.

The authors could report the results of the familiarization and test trials in Experiments 1 and 2, and then if they wished do additional comparisons (e.g., aggressor vs. bystander).

5. If the authors prefer having all 4 main conditions in one analysis, then I would suggest (a) doing a planned comparison for each condition, adjusted for the 4 comparisons; and (b) doing a planned interaction comparison to compare the key conditions two at a time, again with a p level adjusted for the number of such comparisons.

Other points

--Abstract and Introduction: The authors may not like the approach I suggest for framing their research, and what they do is of course up to them. But changes are needed to address the issues raised above (moral traits vs. character; a more straightforward acknowledgment of the fact that the present research builds on and extends prior similar research; and a stronger basis for making predictions about infants' responses to the novel harm scenarios shown here).

Line 89: "Research addressing this question has demonstrated both successes and limitations". As should be clear from the text above, I do not agree that prior reports show particular limitations. They simply present infants with different moral situations than the ones explored here, leading to different inferences and expectations from infants.

Line 131: Prior research on early fairness suggests that younger infants do less well with distributions of 4 items (3:1 violation) as opposed to 2 items (2:0) violation (Buyukozer Dawkins et al., 2019). Are the results different if the authors re-do the present analyses without including infants below 14 or 15 months of age? As things stand, infants in a condition might have shown a weaker preference for the unfair event simply because there happened to be, by chance, more young infants in that condition.

Figure 2 is confusing: It would be better to not have a line connecting all the points vertically for each bar, as that line means something different than the line connecting each infant's two looking times. Separately, did the authors check for ceiling infants (i.e., infants who looked 30 or near 30 s in both test trials)? And for outliers? Were there, by chance, more of these in the victim condition? Do any infants need replacement?

Line 255: "Taken together, our findings demonstrate that infants can make moral trait inferences in multi-agent scenarios containing morally complex agents rather than just dyadic scenarios, as has been standard in previous research (Gill & Sommerville, 2023; Surian et al., 2018; Ting & Baillargeon, 2021). This, in a nutshell, is the main limitation of this paper. It is not particularly novel that infants would be able to reason and draw inferences about interactions among 3 as opposed to 2 agents (we already know this from other work); trying to make that interesting makes for a dull paper and undermines the true interest of the present data.

Line 262: "Second, infants actively formed different expectations for positive and negative characters rather than only

suspending expectations for one of the agents (e.g., Surian et al., 2018; Gill & Sommerville, 2023).” Both Ting and Baillargeon (2021) and Gill and Sommerville (2023) found evidence that went beyond suspending expectations, as noted in the summary above.

Line 264: “Our findings may have differed from prior infant studies because our participants witnessed two contrasting roles in the same interaction” Both Surian et al. (2018) and Ting and Baillargeon (2021) discussed this issue.

Line 312: “Do infants also make trait inferences in non-moral domains or is the moral domain “privileged” as some have suggested (Goodwin et al., 2014). Ting and Baillargeon (2021) also raised this issue at the end of their Discussion and hence might be cited here.

Version 1:

Decision Letter:

Dear Mr Zeng,

Your manuscript titled "Infants Make Moral Character Inferences in Multi-Agent Social Interactions" has now been seen by our reviewers, whose comments appear below. In light of their advice I am delighted to say that we are happy, in principle, to publish a suitably revised version in Communications Psychology.

We therefore invite you to revise your paper one last time to address the remaining concerns of our reviewers and a list of editorial requests. At the same time we ask that you edit your manuscript to comply with our format requirements and to maximise the accessibility and therefore the impact of your work.

Please ensure that you address all presentational concerns through textual revisions and ensure sufficient descriptive data are reported in the text. We appreciate your sharing the code and data, and ask you to make sure these are well documented so that readers with additional exploratory research questions such as those listed by Reviewer #3 can build on your work.

EDITORIAL REQUESTS:

SUBMISSION INFORMATION:

OPEN ACCESS:

Communications Psychology is a fully open access journal. Articles are made freely accessible on publication. For further information about article processing charges, open access funding, and advice and support from Nature Research, please visit <https://www.nature.com/commpsychol/open-access>

* **DATA AVAILABILITY:**

Link Redacted

Best regards,

Jennifer Bellingtier

Jennifer Bellingtier, PhD
Senior Editor
Communications Psychology

REVIEWER EXPERTISE:

Reviewer #1 social cognition, developmental psychology
Reviewer #2 social cognition, developmental psychology
Reviewer #3 social cognition, developmental psychology

REVIEWERS' COMMENTS:

Reviewer #1 (Remarks to the Author):

I am satisfied with the author's response and revisions.

Reviewer #2 (Remarks to the Author):

The authors have satisfactorily addressed all of my original concerns. I recommend acceptance.

Reviewer #3 (Remarks to the Author):

The authors have done a very nice job of addressing many of the issues raised in the review process, and their paper is

much stronger as a result. Below are a few additional issues the authors should consider in preparing the final version of their paper.

Main points:

1. P. 7 and elsewhere: The idea that infants might view victims of harmful or unfair treatment as morally ambiguous, based on research with adults on "victim blaming", seems far-fetched and implausible. There is no evidence with infants to support it, and quite a bit of evidence suggesting the opposite. For example, consider the victims of unfair distributions. Why would infants expect fair distributions in the first place, if they tended to blame disadvantaged victims for their treatment? Why would infants expect leaders to intervene and rectify unfair distributions in their groups? Why would infants evaluate unfair distributors negatively and expect them to be punished for their actions, if disadvantaged victims were somehow morally ambiguous?

And indeed, infants in the present experiments gave no evidence that they viewed the victim as morally ambiguous: They expected the victim (like any other agent of good character) to act fairly toward others, and they were surprised when it did not. In the familiarization trials, it was the aggressor and the cowardly bystander, not the victim, who gave evidence of a poor moral character. Infants received no particular evidence about the victim's moral character, so they applied their general, default expectation that the victim possessed a good moral character, and they assumed it would act fairly.

I would recommend going through the manuscript and revising all statements referring to the victim as morally ambiguous.

2. p. 9 and elsewhere: It is very plausible that having siblings and/or daycare experience would shape infants' sociomoral expectations. Indeed, most developmental researchers would no doubt agree with this suggestion. However, the authors offer an odd rationale for this: They argue that "prior work with adults ties social network complexity with the nature of inferences that adults make after seeing moral exemplars: whereas adults with large social networks typically form broad moral character inferences from moral exemplars (i.e., that someone is a morally good or morally bad person), adults with smaller social networks tend to form more specific moral trait inferences from these same exemplars". As a result they suggest that "infants with siblings and daycare experience will be more likely to make moral character inferences as these infants will (A) have increased social experience giving them more opportunities to learn about others' character, and (B) have larger social networks, and may reap more benefits from representing their social contacts along a single moral character dimension rather than with respect to many specific moral trait dimensions". Describing 12- to 24-month-old infants who have a sibling or attend daycare as having a large, complex social network seems far-fetched to me; the authors' first argument is sufficient to make their point. Specifically, with experience, one would expect infants (a) to become able to make finer discriminations when evaluating others' moral characters (e.g., they might be more likely to view an aggressor as having a vicious moral character), (b) to become generally more efficient at forming moral evaluations of others' characters, or (c) both. Ideas about large social networks do not need to be invoked to make such experiential effects plausible; in fact, they detract from the authors' otherwise well-taken arguments.

3. I found the inclusion of the experiential variables (siblings, daycare) in Experiment 2 a valuable and interesting addition to the paper. However, the data are not sufficiently analyzed, more needs to be done. First, and most importantly, we need to know how the infants with less vs. more experience differed in their responses to the protector and aggressor conditions. There could be many ways for this to happen. For example, the less experienced infants could tend to look equally at all of the events (this would suggest that they needed more familiarization to process the events); or they could show the same longer looking at the unfair vs. fair event in both conditions (this would suggest less discrimination); or they could show longer looking at the unfair event in the protector condition and equal looking at the two events in the aggressor condition, whereas the infants with more experience might show longer looking at the fair event in the aggressor condition. To address this issue, the authors could split their data by some reasonable criterion (or just do a median split) and show us each group's responses.

Second, the authors combine together the sibling and daycare information, but it would be helpful if they could tell us whether both variables mattered equally or whether one variable mattered more than the other.

4. In reporting their main results, the authors focus on how much longer infants looked at the fair than at the unfair event, but that seems to me an odd choice. What the authors find is that in many conditions (protector, victim, random), infants tended to show the typical effect, longer looking at the unfair than at the fair event. This reassures readers that the authors' method was generally sound as it yielded the typical effect. But in addition, the authors show us that in the bystander and aggressor conditions, there were deviations from this typical, expected pattern. From his perspective, it would make more sense to focus on looking at the unfair vs. the fair event, rather than that opposite. Figure 3, in particular, would be more intuitive to readers if it focused on the unfair event, as it says (incorrectly) on the vertical axis.

Other points:

76-80: "By the second year of life, infants expect agents to share resources fairly (i.e., equally) between two recipients (Schmidt & Sommerville, 2011; Sloane et al., 2012)". Actually, infants in the first year of life also expect agents to share resources equally between two recipients: see Meristo et al., 2016 (10 months), and Buyukozer Dawkins et al., 2019 (9 and 4 months). These studies support the claim that an expectation of fairness emerges early in infancy, and they should also be cited.

101: the word infants is repeated.

132-133: "In past studies, negatively valenced agents typically engaged in indirect harm towards recipients, such as hindering their goals (Gill & Sommerville, 2023), treating them unfairly (Surian et al., 2018)". Is it possible that you accidentally reversed these two references?

245-246: "Given that we are using the violation-of-expectation method, we also acknowledge that there is active debate over the validity of these methods (Aslin, 2007; Paulus, 2022)." It might be good to also cite the recent overview of the VOE paradigm by Margoni, Surian, and Baillargeon (2024), which presents the opposite side of this debate.

252: "infants in the aggressor, protector, and victim conditions received identical test trials and highly similar test trials". Identical familiarization trials?

279-280: The video stopped once the aggressor contacted the protector, and a ding sound played to indicate that looking time coding should begin". In experiments 1 and 2, the authors apparently coded infants' looking behavior only during the paused scenes at the end of the trials. In their future research, however, the authors should consider also coding infants' looking behavior during the long (here 32 s) event sequences that precede these paused scenes. It is important for the scientific community to know that infants in the different conditions attended about equally at the events shown in each trial. Moreover, infants who did not attend sufficiently to the events should be eliminated (VOE responses are meaningless if infants did not fully attend to the events).

298: perhaps "two novel recipients, green triangles"?

318-320: "Further analyses are included in the Supplemental Materials to confirm that infants' looking time in the familiarization trials did not significantly affect their looking time in the test trials (Supplemental Methods 2)". There is something a bit odd about the familiarization data reported in the SI. In the cover letter, when comparing the familiarization data in the initial aggressor and protector conditions, the authors found a significant effect of condition. In the SI, when analyzing the familiarization data of the various conditions in Experiment 1, the authors again found an effect of condition, but now they write that "This difference in looking by condition was not particularly surprising given that infants saw different familiarization trials in the Bystander and Random Movement Conditions". So it is unclear exactly what is going on here. Could the authors include a table with the mean familiarization trials in the 5 different conditions? Could they do follow-up analyses to inform their readers about which conditions reliably differed and which did not? And the same would apply to Experiment 2.

453: Infants in the Protector Condition?

464: Ceiling babies are typically those who tend to look (near) the maximum allowed in both test events. We can see in Figure 4 that at least two infants in the aggressor condition of Experiment 2 looked near 30 s at both events; there were no such infants in the same condition of Experiment 1. Could including (as opposed to eliminating) those ceiling babies have contributed to the negative result of the aggressor condition in Experiment 2?

537-540: "Infants' capacity to make inferences concerning morally ambiguous agents also suggest that infants may view others' character along a moral continuum rather than making categorical good vs bad judgements". Ting and Baillargeon (2021) explicitly argue that infants evaluate moral character along a continuum (vicious, bad, good, virtuous) and present data that support such a notion.

545-546: "Our findings suggest that infants, at least in the current context, treat victims as virtuous". In this literature, virtue often means going beyond what is morally required, like helping an outgroup member in need, or giving away a large share of a resource (see Ting & Baillargeon, 2021 for discussion). From this perspective, the victim was not virtuous, as noted above: It simply was assumed to possess a good character (there was no evidence to the contrary) and hence was expected to act fairly. In this scheme, infants in the present experiments assigned different characters to the agents: the aggressor was vicious (for infants who were surprised that it acted fairly), the bystander was bad (infants suspended their expectation that it would act fairly), the victim was good (infants expected it to act fairly), and the protector was presumably virtuous (there was no test of its virtue included in the experiments).

603-605: "By sampling a broad age range that included a range of social experience, the effect in Experiment 2 may have been diluted by increased variability among infants with lower social experience that are in this transitional period.". This explanation for the discrepancy between the aggressor conditions in Experiments 1 and 2 is puzzling given that both experiments used the same age range, 12 to 24 months. Perhaps the authors mean that the larger sample used in Experiment 2 (48 vs. 32) introduced more variability in social experience?

Overview of Changes

Thank you for your detailed comments and feedback. In response to your comments, we made several changes that we believe have greatly improved the strength of our paper.

To begin, to address comments by Reviewer 1 and Reviewer 3, we have clarified that our focus is on infant's capacity to make broad moral character inferences rather than make specific trait inferences, enriched our consideration of background literature, and more accurately situated our results within this prior literature. Additionally, in response to comments from all three reviewers, we have further justified our methods, added additional analyses, and expanded our discussion of the limitations of our work.

Most critically, we conducted a direct replication of two key conditions, the Aggressor and Protector Conditions, with a highly powered sample ($N = 48$ per condition) now included as Experiment 2. This replication enabled us to address some unexpected findings in the familiarization trials between the Aggressor and Protector condition in Experiment 1 (see below), and to more robustly explore the relation between infants' age and their expectations.

This replication also allowed us to address Reviewer 2's comments concerning the validity of infant looking time measures by linking infants looking behaviours to a real world-variable, and, at the same time, address a novel exploratory question concerning the relations between infants' sibling and daycare experience and their character inferences. This was motivated by (A) findings which suggest that infants and toddlers social environment is related to a variety of socio-moral and cognitive capacities (e.g. Burke et al., 2023; Okocha, et al., 2024; Ziv & Sommerville, 2017) and (B) new findings demonstrating that adults with larger social networks, as measured by their day-to-day interaction and online friends, tend to rely more strongly on broad character inferences (believing for example that other's character exists on a single morally good to bad continuum; Jackson et al., 2023)

To preview, Experiment 2 replicated our findings in Experiment 1 which demonstrated that infants fairness expectations for an agent differed based on whether that agent was previously an aggressor or a protector. Our analyses also demonstrated that infants with siblings and daycare experience more strongly differentiated between conditions than infants without siblings or daycare experience, highlighting that infants early social environment plays an important role in their propensity to make moral character inferences.

Thank you, once again, for taking the time and consideration to review our submission. Detailed point-by-point responses to each reviewer are listed below.

Responses to Editor's comments

Editorially, we consider it important that the revised manuscript address the reviewers' concerns regarding the inference of specific moral traits by the participants and better situate the findings within the existing literature. Please note that while you may make revisions to clarify the terminology, we are not asking for a reframing of the study. We encourage you to conduct the additional analyses as suggested by Reviewer 3, however as the study was preregistered, please also report the originally planned analyses (for our policy on preregistration please see: <https://www.nature.com/commspsychol/editorial-policies/preregistration-policy>). Preregistered analyses should not be relegated to the Supplementary Information.

Thank you for this feedback. In our revision, we have made clear throughout the Introduction (and, in particular, on pp. 3, line 47-67) that the focus of our manuscript is on infants' moral character inferences as opposed to specific moral trait inferences. In addition, our enriched discussion of past literature, which now more accurately captures the unique implications of our work, can be found on pp. 5-7, lines 95-148. We have completed the additional analyses requested by Reviewer 3 (see Supplemental Results 1). As required, our pre-registered planned analyses remain in the main text of the manuscript.

Responses to Reviewer 1

However, I am not convinced that the authors can demonstrate trait inference from the results of the experiments, especially the negative one (aggressor condition). In the aggressor condition, the prior role of the agents is aggressor, which is attributed to physical traits such as strong or violent, rather than the moral trait. So I am not sure about the link between the aggressor and the unfair distributor. How can the authors explain the link between them in terms of the moral trait?

We agree that the link between aggressive behavior and unfair behaviour is not fully articulated in the manuscript.

To clarify, both behaviors can reflect morally relevant traits: aggressive behavior might reflect a lack of concern for other's welfare (i.e. uncaring or hurtful) and unfair behaviour might reflect a lack of concern for principles of justice (i.e. unjust). We agree that while these traits are not identical, they are both indicative of poor moral character. Critically, research suggests adults *do* link the domains of physical harm and unfairness (Landy & Bartels, 2018), given their common indication. Moreover, research with adults suggests that global positive/negative moral character inferences predominate over more specific moral trait inferences in impression formation (Pringle et al., 2023), making inferences about moral character (rather than the attribution of specific moral traits) a particularly important area of research. Accordingly, we have reoriented our introduction to focus on broad moral character inferences (e.g. good vs bad) rather than inferences of specific moral traits (e.g. violent, unjust, hurtful). Changes have been made throughout the Introduction and Discussion, and see in particular pp. 3, lines 63-67 and pp. 8, line 171.

Related to this point, if the authors can explain the link, they could discuss why they found the negative trait inference, which has not been found in previous studies (Surian et al., 2018; Gill & Sommerville, 2023).

One reason infants may have formed expectations for unfairness in our study but not in other studies may be because the initial immoral behaviour we presented infants with was more severe than in other studies.

In our study, the aggressor physically harmed another agent, hitting them or squishing them against a wall 14 times per familiarization video. In contrast, in other studies, infants observed

agents destroying the possessions of another agent (Ting et al., 2021) or hindering another agent from achieving a goal but not physically harming them (Gill & Sommerville, 2023) only a single or a few times per familiarization video. In addition to showing more occurrences of the immoral behavior, research suggests that physical harm may be seen as more egregious than other moral violations. In adults, research suggests that harm is the primary mode through which moral judgements are made (Schein & Gray, 2022). In children, numerous studies have suggested that physical harm is judged more harshly than unfairness, emotional harm, and damage to physical property (e.g. Yucel et al., 2022; Smetana & Ball, 2019; Elkind & Dabek, 1977). Thus, we speculate that our use of many physical harm-related actions led infants to make stronger negative moral character inferences of these agents, leading to greater expectations of future moral transgressions (Hartman et al., 2022).

We have articulated this point in the Introduction (pp. 6, line 130-138).

Another concern is the wide age range of the participants. The authors should justify in the manuscript why they chose such a wide range of ages.

We have included further justification for the age range in the Introduction (pp. 7, line 150-157). In sum, we chose a large age range so that we could explore the effects of age on infants' character inferences and because infants possess the necessary prerequisites for making moral character inferences by the age of 12 months (basic understanding of agents, protective third-party intervention, and fairness; Woodward et al., 1998; Kanakogi et al., 2017; Ziv & Sommerville, 2017).

Minor points;

-Line 82, the authors misunderstand the findings of Kanakogi et al. (2017).

Thank you for noticing this error. We have corrected the text in lines 87-89 to accurately reflect experiments 4 and 5 of Kanakogi et al., 2017.

-Line 98-101, I am not sure about the first limitation. Please add the example.

We have removed the comment framing this as a limitation, as we believe that it was slightly misleading. Instead, we have focused on highlighting the major finding from this general body of work, which is that at least under some circumstances, infants can make moral character inferences (Introduction, pp. 5, lines 95-109).

-The authors have overlooked the study by Kanakogi et al. (2013) in their reference.

We have added Kanakogi et al. (2013) to our references and included a citation on line 127.

-Line 375-377, why did the authors analyse the violation of expectations in the familiarisation phase?

We analyzed the familiarization phase to ensure that looking time in test trials was not driven by looking time in the familiarization trials. In this version of the manuscript, we have moved these

analyses to the Supplemental Materials (Supplemental Method 2) and clarified the reasoning for why we conducted these analyses.

Responses to Reviewer 2

However, there are some aspects to consider. The use of abstract geometric shapes as agents, while allowing for precise control of stimuli, could constrain the validity. Most, but not all previous studies with infants and toddlers have used human actors in test stimuli. The authors should elaborate on how their approach using geometric shapes differs from these prior methods and discuss potential implications for interpreting the results.

We have included further justification for our choice to use abstract geometric shapes over human actors in the Method (pp. 12-13, lines 259-267).

In sum, we choose to use abstract shapes as these types of stimuli have been used in a variety of studies examining infants' early socio-moral development including the domains of helping (Hamlin et al., 2007), physical harm (Kanakogi et al., 2013), and fairness (Surian et al., 2011). Furthermore, evidence suggests that findings relying on puppets/animated agents and findings relying on human actors often converge (Kominsky et al., 2022). For example, infants expect both human actors and animated agents to act in a goal directed and efficient manner (Woodward et al., 1998; Liu et al., 2019; Adam et al., 2019; Gergely et al., 2005). In the domain of fairness, Schmidt & Sommerville (2011), who use human actors, and Surian et al. (2011), who use animated shapes both demonstrate that infants expect agents to distribute resources fairly rather than unfairly. More directly related to our research, in past investigations of infant's moral character inferences, infants expect helpers to behave fairly but not hinderers regardless of whether the helping and hindering agents are animated shapes or human actors (Surian et al., 2018; Gill & Sommerville, 2024).

Together, these findings suggest that animated geometric shapes yield perceptions of agency and thus can be used as a proxy for human actors, while allowing for greater control over critical stimuli dimensions.

Additionally, the reliance on looking time as the only dependent measure, while well-established in infant research, has limitations. The authors should more thoroughly discuss the validity of using looking time to infer infants' moral expectations and consider potential alternative explanations for the observed patterns.

We acknowledge the concerns that have been raised regarding looking time methods, and we have included additional discussion of this topic in the Method (pp. 21, line 409-420). While it is unlikely that our findings are subject to lower-level perceptual expectations, given that infants in the aggressor, protector, and victim conditions receive identical familiarization and nearly identical test trials, we sought to conduct a replication study with an ecologically valid correlate to directly address these concerns.

In Experiment 2, we replicated the Protector and Aggressor Condition and measured infants' social contact – namely their sibling status and daycare experience. As described on pp. 24-25,

line 467-494, in Experiment 2 we replicated the key condition by test trial type interaction and also found that infants' differentiation between moral roles is related to their amount of social contact (more social contact was associated with greater moral role differentiation). Thus, infants' expectations in this study had some basis in their real-world experience, consistent with past work showing that infants' performance on violation-of-expectation paradigms tracks their everyday experience (e.g. action understanding; Sommerville & Woodward, 2005)

Furthermore, the authors could situate their findings in relation to previous studies on infants' and toddlers' fairness reasoning, examining how the lack of age effects aligns with or differs from prior work.

One challenge with interpreting age effects in the context of previous studies is that past investigations of infant's moral trait inferences have either had narrow age ranges, did not report age analyses, or were underpowered to detect age effects (Surian et al., Gill & Sommerville, 2024; Ting & Baillargeon, 2021). Thus, although these studies characterize age ranges in which infants can make moral character inferences, they cannot speak to age related changes in infants' capacities.

An advantage of our approach was that across both of our experiments, we were much better powered to detect age effects, particularly in the Aggressor and Protector Conditions (n = 80 per condition across two experiments). Overall, we found that age was unrelated to infants' expectations, with the exception that infants' expectations in the aggressor condition were weakly and marginally related to age. Experiment 2 revealed that infant's moral character inferences were significantly predicted by their sibling and daycare experience. Furthermore, when age and sibling/daycare experience are included in the same model, the effect of sibling and daycare experience remains significant, and the effect of age continues to be non-significant. These findings raise the possibility that infants moral character inferences are more strongly driven by the increases in social contact and experience that come with age rather than age itself.

See pp. 26-27, line 496-516 for these results, and pp. 30-31, line 574-598 in the Discussion for commentary on this topic.

Finally, the discussion section would benefit from an expanded consideration of the universality versus cultural specificity of these moral trait inferences. Given that moral values and judgments can vary significantly across societies, it is crucial to address whether the observed abilities in infants from a Western cultural context would generalize to infants from different cultural backgrounds.

We agree with the reviewer that considering universality and cultural specificity is important. We have included additional discussion (pp. 31, line 608-615) of cultural differences in moral development in infancy citing research suggesting that infants' fairness expectations are not identical across cultures (Meristo & Zeidler, 2022) and cautioned against universalizing interpretations of our findings.

Responses to Reviewer 3

1. One issue has to do with the framing used in the Abstract and Introduction, which does not serve the research well. First, it is not clear to me that the present research has to do with infants' ability to infer specific moral traits (e.g., honesty, kindness, humility, courage, and so on); a broader focus on moral character would, I think, serve the research better. I may be wrong, but I suspect that many readers would view an agent's moral character less as a specific moral trait and more as a global assessment of the agent's moral disposition.

We agree with the reviewer that an investigation of the ability to make moral character inferences more accurately characterizes our research. We have reoriented our manuscript around broad moral character inferences and cited literature which suggest that broad moral character information takes precedence over more specific character information in adults impression formation to motivate this decision (Pringle et al., 2023; pp. 3, line 63-67).

Second, the authors need to be more straightforward in presenting prior reports, both to acknowledge their contributions and to make clear what is novel about the present research. Going chronologically, Surian et al. (2018) found that 15-month-olds who first saw A help B achieve a goal (in 2 familiarization trials) and saw C hinder B from achieving this goal (in 2 familiarization trials) were later surprised if A divided resources unfairly between two novel targets (2:0), but looked equally whether C acted fairly or unfairly toward the novel targets. In a similar vein, Ting and Baillargeon (2021) found that after seeing A harm an ingroup's or an outgroup's property in 3 familiarization trials, 25-month-olds looked equally if A next divided resources unfairly between two ingroup targets (2:0), but were surprised if A chose to share resources generously with an ingroup target. Finally, Gill and Sommerville (2023) replicated the findings of Surian et al. with 14- to 27-month-olds, using a 5:1 violation. In addition, they found that when the events shown in familiarization and test were reversed, with A acting fairly and C unfairly, infants looked equally in test whether A chose to help or to hinder B (since there was no indication that A and B belonged to the same group, either action was acceptable), whereas infants looked reliably longer when C chose to help as opposed to hinder B (helping an outgroup is a supererogatory, virtuous action that one would not expect from an agent with a poor moral character).

Together, these results suggest that when infants take an agent's negative action to provide evidence of a poor moral character (see Ting & Baillargeon, 2021, for discussion of some of the factors that affects such decisions), they then use this inference to predict the agent's actions in new contexts. For example, they are not surprised if an agent who gives evidence of a bad moral character later acts unfairly (the default expectation that agents will be of good character and will act fairly is weakened). Conversely, they are surprised if an agent who gives evidence of a bad moral character later acts virtuously (i.e., goes beyond what is obligated by a good character), for example, by sharing resources generously with an ingroup member, or by kindly helping an individual who is not clearly identified as an ingroup member.

In contrast, when agents' actions do not lead infants to question the goodness of their characters, then typical expectations apply (e.g., agents will act fairly, and they may choose to help or hinder outgroup members as they wish—helping is obligatory only for ingroup members).

We appreciate the detailed comments that the reviewer has provided. We agree that our discussion of past work was insufficient and have included much more detail in the Introduction on pp. 5-6 from lines 108-121. The main change that we have made here includes providing a more comprehensive overview of each of the findings in these three studies in order to more clearly articulate the specific contribution of our work, which is the consideration of a more complex social interaction that contains multiple morally ambiguous roles.

Sorry for the long review—it helps situate the present work. What is exciting here is that the authors raise two novel questions that go beyond these prior findings. First, what happens when an agent gives evidence of having not simply a bad character, but a vicious character? This is an issue that was raised by Ting and Baillargeon (2021) in their discussion: “What if the wrongdoer first committed a more severe violation, by directing more harmful actions at the victim or by harming a larger number of victims? Would children now expect her to act unfairly in the test trials, and hence would they look significantly longer if she acted fairly instead?” (p. 9). In the present research, the harm is directed at the victim itself (i.e., at its body), and it is repeated numerous times. In the prior research, infants and toddlers saw only 2 or 3 instances of hindering or harming (destruction of personal property) to start. But here we have an aggressor who does great bodily harm to a victim: Each familiarization trial lasts 30 s and involves 11 or 12 hits plus 3 squishes! Across the 4 familiarization trials, this represents a great deal of bodily harm (over 40 hits plus 12 squishes), leading infants to attribute a very bad (or vicious) moral character to the aggressor.

Second, what happens when an agent does not commit such harm but allows it to happen without ever intervening? Indeed, the “bystander” continually avoids the aggressor by staying as far away from it as possible, and it eventually comes out on the opposite side of the cage. This all conveys a defensive and perhaps cowardly posture that, to my mind at least, makes the term “bystander” less than adequate. This is not just a passive bystander who observes what is going on; this is a character who appears to fear the aggressor and stays away from it, in trial after trial. Indeed, the results suggest that infants attribute a poor character to the cowardly bystander, who is guilty of harm by omission.

We agree with the reviewer that more attention should be given to the specific actions of each agent, particularly the aggressor and the bystander. For the aggressor, we have included additional discussion in the Introduction (pp. 6, lines 130-138) to contextualize how the aggressor’s action (physical harm) differs from agents in past research that only engage in hindering, unfairness, or property destruction. We cite research demonstrating that physical harm is often judged more negatively than other forms of harm (e.g. Yucel et al., 2022; Smetana & Ball, 2019; Elkind & Dabek, 1977) and argue that this may lead infants to form heightened negative moral character inferences of the aggressor (mirroring and citing Ting & Baillargeon, 2021).

For the bystander, we have cited additional work suggesting that adults negatively evaluate moral inaction, but that this role still contains some aspects of moral ambiguity given that moral inaction is not judged as harshly as direct harm. (Spranca et al., 1991; pp. 7, line 144-145). In the

Discussion (pp. 28-29, lines 547-554), we further discuss how the moral inaction and the cowardly nature of the bystander may both contribute to infants' expectations.

The approach I am suggesting echoes some of the themes the authors lay out in the Discussion. Embracing this approach from the Introduction would have two important benefits: It would make clear how the present research builds on and extends prior findings, and it would allow the authors to make richer and more interesting predictions about the likely results in each condition. As things stand now, it is unclear why the authors predict what they do in the aggressor condition (vicious character), the bystander condition (poor character), the protector condition (virtuous character), and the victim condition (good character).

Although we agree with the points the reviewer has made, we would like to clarify that we did not preregister directional predictions for the bystander and victim conditions (as stated in pp. 8, lines 172-174 of the Introduction).

We chose not to preregister directional predictions for these two agents given that their moral character is more ambiguous than that of protectors and interveners. Specifically, we cite research suggesting that immoral inaction is evaluated less harshly than immoral action (Spranca et al., 1991), and that adults often have diverging judgements of victims, sometimes viewing them as virtuous victims (Jordan & Kouchaki, 2021), but also sometimes engaging in victim blaming (Niemi & Young, 2016). Discussion of this point can be found the Introduction (pp. 7, line 139-145) and Discussion (pp. 28-29, line 542-554).

We further clarify in the Introduction and Discussion (pp. 5, line 113-115, pp.28, line 540), that as adults, we view each of these characters along a moral continuum, and that it is possible that infants do so as well.

2. I would suggest more standard analyses of the data that are less likely to raise questions about corrections for multiple comparisons. For example, the authors could have two experiments, one focusing on the aggressor (severe bodily harm by commission) and one on the bystander (severe bodily harm by omission). Experiment 1 would include the aggressor condition (fair > unfair) and the protector condition (unfair > fair, the standard result). After showing that the familiarization data are similar in the two conditions (to make sure these are not directly responsible for any test differences), the authors could focus on the test data, where we would expect a significant interaction between the two conditions. Experiment 2 could have the bystander condition (unfair = fair) and victim condition (unfair > fair), with again a significant interaction between the two conditions. This way, in each experiment, one condition would show the standard result (unfair > fair), and one condition would deviate from it, as predicted.

While we agree that analyzing the data in this suggested manner (i.e., splitting the data into Experiments 1 and 2, as suggested above) would be a useful way to gain clarity on our findings, given the pre-registration, journal guidelines, and the editor's comments (pp. 1-2 of this document), we have retained our pre-registered analyses in the main text. Nevertheless, we performed these suggested analyses, and they are presented below. As they reveal, the exact anticipated findings on test trials suggested by the reviewer were obtained. We have also added this analytic approach to the Supplemental Materials (Supplemental Results 1).

We also found, using this new analytic approach, some puzzling results in the familiarization trials between Aggressor and Protector Conditions. Namely, infants looking differed between the familiarization trials in these two conditions, despite infants being shown the exact same familiarization videos. Critically, follow up analyses revealed that these differences in familiarization looking cannot account for the between-condition differences in the test trials.

Given this unexpected finding (and other reasons listed in pp. 1 and in response to other reviewers' comments, e.g. pp. 4-5 of this document), we decided that the most rigorous approach would be to conduct a pre-registered replication of the Aggressor and Protector conditions in Experiment 2. Here, we replicated the interaction between the Aggressor and Protector condition on test trials and found equivalent looking between conditions on familiarization trials (see pp. 22-23, lines 447-443 and Supplemental Methods 2). This replication therefore ensures that differences between the Aggressor and Protector condition are not due to differences in familiarization looking and overall adds robustness to the findings.

Finally, we agree with the reviewers' comment that multiple comparisons are a concern. Because of this, we have included adjusted p-values using the Tukey honest significant differences (HSD) method in the main text of our manuscript (Table 1, pp. 13-14). More details about these analyses are included in responses to the subsequent comments

Reviewer's suggested re-analyses

“Experiment 1”: Comparing Aggressor and Protector Conditions

To determine whether infants' expectations differed across the Aggressor and Protector conditions we conducted a mixed ANOVA with condition as a between-subjects factor, test trial type as a within-subjects factor, and looking time as the dependent variable. This analysis demonstrated that there was a significant test trial type by condition interaction ($F(1, 62) = 9.03$, $p = .004$), and no significant main effects of test trial type or condition. Thus, as predicted, a significant interaction between conditions was obtained.

“Experiment 2”: Comparing Bystander and Victim Conditions

We sought to determine whether the Bystander and Victim Conditions differed from one another. A mixed ANOVA demonstrated that there was a significant test trial type by condition interaction ($F(1,62) = 4.47$, $p = .029$), and no significant main effects of trial type or condition. Thus, as predicted, a significant interaction was obtained.

“Experiment 1”: Familiarization analyses

To ensure that familiarization events were attended to equally across conditions, we conducted a mixed ANOVA specifying condition as a between-subjects factor, familiarization number as a within-subjects factor, and looking time as the dependent variable. There was a main effect of condition ($F(1, 62) = 7.8$, $p = .006$). There was also a main effect of familiarization number such that infants looking decreased in each subsequent test trial ($F(3, 186) = 10.60$, $p < .001$), and no significant test trial and familiarization number interaction.

The main effect of condition in these analyses was surprising given that infants saw the same video in both conditions and introduces the possibility that between condition differences may have been driven by differences in attention towards the familiarization trials. To ensure that these effects were not driven by differences in attention to the familiarization videos, we conducted a linear regression specifying condition and total looking time towards familiarization as predictors, and difference in looking time as the dependent variable. These analyses demonstrated that there was no significant effect of total looking time towards familiarization on the test trials ($t(62) = .97, p = .33$), and that the effect of condition remained significant ($t(62) = 4.1, p < .001$). The analysis conducted with proportion scores found similar results, and total looking also did not have any significant effects when condition was removed from the model.

“Experiment 2”: Familiarization analyses

A mixed ANOVA examining the familiarization trials demonstrated that there was a main effect of familiarization number ($F(3, 186) = 6.76, p < .001$) such that infants looking decreased on each subsequent trial, but no main effect of condition or interaction effect.

3. I think the random condition is problematic and should be relegated to the SI. It is not clear what it is controlling for, and it did not yield the expected result of longer looking at the unfair as opposed to the fair event. This suggests that infants were puzzled by the random movements of the characters (what are they doing??) in the 4 familiarization trials, leaving them to suspend their expectations in the test trials.

Since three different looking-time patterns are found in the other four conditions, in a sensible and predictable way, the random control adds little to the conclusions. Moving that control condition to the SI would also help limit the number of comparisons performed.

Unfortunately, as this study was pre-registered, we are not able to move the random condition to the Supplemental Information (see editors’ comments on pp. 1-2 of this manuscript for more details on journal policy). However, we agree that the random condition was not an ideal control condition, and we have included additional discussion regarding this condition in the Discussion (pp. 28, line 531-534), noting in particular that infants expect agent to act in a manner that is goal directed and efficient (Woodward et al., 1998; Liu & Spelke, 2017), thus making random movement relatively atypical behaviour for an agent. The analytic approach that was suggested in the prior comment, and that we have included in the supplemental materials also does not contain this condition.

4. For me, the current analyses, as reported, are problematic. First, as noted, there is no control for multiple comparisons within the same overall analysis, something we might expect with so many comparisons. Second, the analyses of the proportion data are non-standard and are potentially unwarranted (they could be deleted). For example, higher levels of looking might be expected overall in the protector condition, which depicts a virtuous, heroic character, than in the bystander condition, which depicts a cowardly character. The simpler and more standard analyses I suggested above would make these proportion analyses unnecessary. The authors could report the results of the familiarization and test trials in Experiments 1 and 2, and then if they wished do additional comparisons (e.g., aggressor vs. bystander).

We have included Tukey HSD corrected p-values in the exploratory analyses (pp. 21-22, lines 435-436). These corrected p-values generally align with the uncorrected p-values exception of a single comparison which becomes null after the correction (Bystander vs Random Movement; $p = .016 \rightarrow p = .078$).

We recognize the reviewers point regarding the proportion analyses. Many violation-of-expectation studies examine the proportion of infants that look longer towards unexpected event (e.g. Woodward et al., 1998). The use of proportion scores as we have done in our exploratory analyses is also common (e.g. Ziv & Sommerville, 2017). Both of these options have been recommended by some scholars as they serve to A) provide a convergent measure, B) help control for outliers, and in the case of proportion scores, C) control for total looking time towards test trials. We have included these points as justification for our approach in the manuscript on pp. 17, line 376 and pp. 20, line 427.

5. If the authors prefer having all 4 main conditions in one analysis, then I would suggest (a) doing a planned comparison for each condition, adjusted for the 4 comparisons; and (b) doing a planned interaction comparison to compare the key conditions two at a time, again with a p level adjusted for the number of such comparisons.

Regarding point (a): We have decided to not correct for multiple comparisons in our main analyses as these were pre-registered, planned analyses. Additionally, each of these comparisons are conducted on independent sets of participants (i.e. each between-subjects condition only included one t-test). More generally, we note that by there are trade-offs to correcting for multiple comparisons. Although this practice reduces the probability of false positives, it also increases the probability of false negatives. Additionally, the decision to correct for multiple comparisons can often be arbitrary (for example, if we had published each of our conditions in separate papers, we would not need to include corrections for multiple comparisons). Indeed, some scholars have argued that this this practice is not necessary, and the instead p-values should be reported as is with the interpretation left to the reader, who is assumed to be scientifically literate, and can consider problems of multiple testing for themselves (Althouse, 2016; Rothman, 1990).

Despite these arguments, we have still chosen to correct for multiple comparisons in the exploratory analyses as we were interested in interpreting these findings and we deemed the added caution appropriate given their exploratory nature. We have also added confidence intervals for most of our analyses as an additional method to aid readers in making informed interpretations of the strength (or lack of strength) of our findings. See pp. 21-22, lines 435-436.

Regarding point (b), we do the equivalent of these analyses in the exploratory analyses using proportion scores/different scores and t-tests. We have included the analyses that the reviewer has recommended in the Supplemental Materials (Supplemental Results 2), and we find that they provide the same results as our exploratory analyses. We have decided to report the proportion scores in the main analyses given the reasons listed previous comments, and because these analyses are simpler to parse.

Other points

--Abstract and Introduction: The authors may not like the approach I suggest for framing their research, and what they do is of course up to them. But changes are needed to address the issues raised above (moral traits vs. character; a more straightforward acknowledgment of the fact that the present research builds on and extends prior similar research; and a stronger basis for making predictions about infants' responses to the novel harm scenarios shown here).

We agree with the reviewer and have attempted to address these issues. To summarize the changes again, we have reoriented our manuscript around the inference of broad moral character rather than the inference of specific moral traits (Introduction pp. 3, line 63-67 and throughout). Additionally, we have included a more detailed discussion of past research on infants' moral character inferences and added theoretical basis for the differences between the agents in our paradigm and the agents used in past paradigms. Particularly, aggressors in our study cause repeated physical harm as opposed to immoral agents in past studies which involve hindering or property destruction, and no previous studies have investigated infants' character inferences of victims and bystanders (Introduction pp. 6-7, lines 130-148).

Line 89: "Research addressing this question has demonstrated both successes and limitations". As should be clear from the text above, I do not agree that prior reports show particular limitations. They simply present infants with different moral situations than the ones explored here, leading to different inferences and expectations from infants.

We did not intend to use limitations in a negative sense here, and we have removed this framing in the paper. Instead, we simply wanted to indicate that in some cases, infants do not make full reversal of expectations (e.g., suspending fairness expectations for hinderers but not expecting them to be unfair; not expecting fair agents to help nor hinder). We acknowledge that in other cases they do make this full reversal (e.g. expecting unfair agents to hinder; being surprised when unfair agents behave generously). These changes can be found in the introduction on pp. 5-6, lines 108-121).

Line 131: Prior research on early fairness suggests that younger infants do less well with distributions of 4 items (3:1 violation) as opposed to 2 items (2:0) violation (Buyukozer Dawkins et al., 2019). Are the results different if the authors re-do the present analyses without including infants below 14 or 15 months of age? As things stand, infants in a condition might have shown a weaker preference for the unfair event simply because there happened to be, by chance, more young infants in that condition.

We did not re-analyze the data excluding kids below 14 to 15 months given that doing so would significantly reduce our power and given that this approach would yield slightly unequal sample sizes by condition. Furthermore, our age analyses (see pp. 26-27, lines 497-506), provided no evidence of significant age effects in any of our conditions.

We also ensured that the average age of infants did not vary by condition in either of our two experiments (Experiment 1: $F(4,155) = .86, p = .49$; Experiment 2: $F(1,94) = .025, p = .87$), and that it did not differ between experiments ($t(188) = .26, p = .80$). To ensure that age distributions

did not differ between conditions, we conducted a k-sample Anderson-Darling test, which tests for whether multiple samples come from the population distribution. This test was not significant in either experiment (Experiment 1: $A = 5.46$, $p = .15$; Experiment 2: $A = 1.31$, $p = .24$), confirming that the age distribution of the sample populations did not significantly differ between conditions.

Figure 2 is confusing: It would be better to not have a line connecting all the points vertically for each bar, as that line means something different than the line connecting each infant's two looking times.

We have removed the vertical lines from the box plot in Figure 2 for greater clarity.

Separately, did the authors check for ceiling infants (i.e., infants who looked 30 or near 30 s in both test trials)? And for outliers? Were there, by chance, more of these in the victim condition? Do any infants need replacement?

We checked for ceiling effects, which we categorized as infants who looked for longer than 29.5 seconds, but they were rare and distributed throughout our sample (Experiment 1: Aggressor - 1, Protector - 4, Bystander - 0, Victim - 2, Control - 1; Experiment 2: Aggressor - 4, Protector - 1). Additionally, there was no change in results if these infants were removed for any of our analyses.

Regarding outliers, we preregistered that infants were considered outliers if they're looking was 3 standard deviations above or below the mean for a test trial. There were no outliers under this criterion.

Line 255: "Taken together, our findings demonstrate that infants can make moral trait inferences in multi-agent scenarios containing morally complex agents rather than just dyadic scenarios, as has been standard in previous research (Gill & Sommerville, 2023; Surian et al., 2018; Ting & Baillargeon, 2021). This, in a nutshell, is the main limitation of this paper. It is not particularly novel that infants would be able to reason and draw inferences about interactions among 3 as opposed to 2 agents (we already know this from other work); trying to make that interesting makes for a dull paper and undermines the true interest of the present data.

We have slightly edited this sentence to emphasize that the novel contribution of this work is not only that infants can make inferences in multi-agent scenarios, but they also do so for agents with distinct roles that exist on a moral continuum (Discussion, pp. 28, line 535-541). As we previously discussed in more detail in another comment (e.g. pp. 7, this document), we also included additional discussion of how the actions in our paradigm differed from past studies, particularly the fact that our study employed physical harm, and that previous studies have not examined infants' expectations towards bystanders and victims (Introduction pp. 6-7, line 130-148; Discussion, pp. 28-29, line 542-554).

Line 262: "Second, infants actively formed different expectations for positive and negative characters rather than only suspending expectations for one of the agents (e.g., Surian et al.,

2018; Gill & Sommerville, 2023).” Both Ting and Baillargeon (2021) and Gill and Sommerville (2023) found evidence that went beyond suspending expectations, as noted in the summary above.

We have decided to remove this point from our discussion given that it only provides limited novelty over other studies, and so that we could increase our discussion of other effects that were introduced in the replication study (i.e. individual difference effects).

Line 264: “Our findings may have differed from prior infant studies because our participants witnessed two contrasting roles in the same interaction” Both Surian et al. (2018) and Ting and Baillargeon (2021) discussed this issue.

We have removed this point from our discussion given that we are no longer centering the reversal in fairness expectations that we found for aggressors as strongly.

Line 312: “Do infants also make trait inferences in non-moral domains or is the moral domain “privileged” as some have suggested (Goodwin et al., 2014). Ting and Baillargeon (2021) also raised this issue at the end of their Discussion and hence might be cited here.

We have included a citation to Ting and Baillargeon (2021) on pp. 32, line 619 for this question.

Reviewer 1

I am satisfied with the author's response and revisions.

Reviewer 2

The authors have satisfactorily addressed all of my original concerns. I recommend acceptance.

Response to Reviewer 3's Comments

The authors have done a very nice job of addressing many of the issues raised in the review process, and their paper is much stronger as a result. Below are a few additional issues the authors should consider in preparing the final version of their paper.

Main points:

1. P. 7 and elsewhere: The idea that infants might view victims of harmful or unfair treatment as morally ambiguous, based on research with adults on "victim blaming", seems far-fetched and implausible. There is no evidence with infants to support it, and quite a bit of evidence suggesting the opposite. For example, consider the victims of unfair distributions. Why would infants expect fair distributions in the first place, if they tended to blame disadvantaged victims for their treatment? Why would infants expect leaders to intervene and rectify unfair distributions in their groups? Why would infants evaluate unfair distributors negatively and expect them to be punished for their actions, if disadvantaged victims were somehow morally ambiguous?

And indeed, infants in the present experiments gave no evidence that they viewed the victim as morally ambiguous: They expected the victim (like any other agent of good character) to act fairly toward others, and they were surprised when it did not. In the familiarization trials, it was the aggressor and the cowardly bystander, not the victim, who gave evidence of a poor moral character. Infants received no particular evidence about the victim's moral character, so they applied their general, default expectation that the victim possessed a good moral character, and they assumed it would act fairly.

I would recommend going through the manuscript and revising all statements referring to the victim as morally ambiguous.

Although we recognize the reviewers point, we believe that the articles that the reviewers cite relate more to infant's expectations and knowledge of moral norms, but do not directly provide evidence regarding infants' feelings towards victims.

We do, however, agree that more justification for the ambiguity of victims could be provided, and thus we cite additional work with infants which suggests that infants' evaluations of victims can be inconsistent or ambiguous (lines 131-136).

Particularly, we cite work from Kanakogi et al. (2013) suggesting that infants preferentially reach towards victims over neutral agents, suggesting that they evaluate victims positively. In contrast, we cite research indicating that infants prefer and help rich over poor agents, and that they expect resource distributions to align with dominance hierarchies (Eason & Enright et al., 2024; Enright et al., 2017). These findings suggest infants' evaluations of the victims of inequality may be negative.

Overall, these findings suggest that infants' evaluations of victims are not so clear cut and motivate the claim that victims may be viewed as morally ambiguous.

2. p. 9 and elsewhere: It is very plausible that having siblings and/or daycare experience would shape infants' sociomoral expectations. Indeed, most developmental researchers would no doubt agree with this suggestion. However, the authors offer an odd rationale for this: They argue that "prior work with adults ties social network complexity with the nature of inferences that adults make after seeing moral exemplars: whereas adults with large social networks typically form broad moral character inferences from moral exemplars (i.e., that someone is a morally good or morally bad person), adults with smaller social networks tend to form more specific moral trait inferences from these same exemplars". As a result they suggest that "infants with siblings and daycare experience will be more likely to make moral character inferences as these infants will (A) have increased social experience giving them more opportunities to learn about others' character, and (B) have larger social networks, and may reap more benefits from representing their social contacts along a single moral character dimension rather than with respect to many specific moral trait dimensions". Describing 12- to 24-month-old infants who have a sibling or attend daycare as having a large, complex social network seems far-fetched to me; the authors' first argument is sufficient to make their point. Specifically, with experience, one would expect infants (a) to become able to make finer discriminations when evaluating others' moral characters (e.g., they might be more likely to view an aggressor as having a vicious moral character), (b) to become generally more efficient at forming moral evaluations of others' characters, or (c) both. Ideas about large social networks do not need to be invoked to make such experiential effects plausible; in fact, they detract from the authors' otherwise well-taken arguments.

We agree that this point may be overly speculative for the introduction of our manuscript, and thus we have removed it from the introduction to keep the focus on past research demonstrating that social experience plays an important role in infants' social and cognitive development (lines 170-179).

3. I found the inclusion of the experiential variables (siblings, daycare) in Experiment 2 a valuable and interesting addition to the paper. However, the data are not sufficiently analyzed, more needs to be done. First, and most importantly, we need to know how the infants with less vs. more experience differed in their responses to the protector and aggressor conditions. There could be many ways for this to happen. For example, the less experienced infants could tend to look equally at all of the events (this would suggest that they needed more familiarization to process the events); or they could show the same longer looking at the unfair vs. fair event in both conditions (this would suggest less discrimination); or they could show longer looking at the unfair event in the protector condition and equal looking at the two events in the aggressor condition, whereas the infants with more experience might show longer looking at the fair event in the aggressor condition. To address this issue, the authors could split their data by some reasonable criterion (or just do a median split) and show us each group's responses.

Second, the authors combine together the sibling and daycare information, but it would be helpful if they could tell us whether both variables mattered equally or whether one variable mattered more than the other.

To examine the effect of social experience in each condition, we conducted separate regressions in each condition (lines 463-467). This analysis demonstrates that in the Protector Condition, infants' looking was significantly predicted by daycare/sibling score ($\beta = .08$, 95% CI [.01, .15], $se = .03$, $t(42) = 2.36$, $p = .023$). In contrast, infants' looking was not significantly predicted by daycare/sibling score in the Aggressor Condition ($\beta = -.03$, 95% CI [-.11, .04], $se = .04$, $t(40) = .91$, $p = .37$).

We chose to use this analysis rather than the similar one suggested by the reviewer because the analysis that the reviewer suggested (i.e., median split analysis) would require us to conduct more statistical tests and result in uneven and smaller sample sizes, which can reduce power. Furthermore, conducting a median split in our data is challenging given that a large proportion of our sample is clustered around the midpoint of 1 for the sibling/daycare experience variable. The analysis requested by the reviewer is presented here: In the Protector Condition, infants with a daycare/sibling score less than 1 did not significantly differ in their looking ($t(16) = 1.09$, $p = .29$, $d = .27$, 95% CI [-.22, .75]), but infants with a daycare/sibling score greater than 1 did ($t(26) = 2.50$, $p = .019$, $d = .48$, 95% CI [.08, .88]). In the Aggressor Condition, infants with a daycare/sibling score less than 1 did not significantly differ in their looking ($t(16) = -.33$, $p = .75$, $d = -.08$, 95% CI [-.55, .40]), nor did infants with a daycare/sibling score greater than 1 ($t(26) = -$

1.27, $p = .22$, $d = -.25$, 95% CI [-.65, .15]).

For the second set of analyses the reviewer suggested, we conducted a linear regression including the interaction between condition and sibling presence, and the interaction between condition and daycare experience as separate variables (lines 456-462). Infants looking was not significantly predicted by the interaction between condition and sibling presence ($\beta = .13$, 95% CI [-.03, .29], $se = .08$, $t(80) = 1.66$, $p = .10$), nor by the interaction between condition and daycare experience ($\beta = .10$, 95% CI [-.06, .26], $se = .08$, $t(80) = 1.28$, $p = .20$), though both these analyses trended in the expected direction. One possibility for these results is that each of these variables contributed small effects that we were underpowered to detect, and it is only by examining their additive impact that we were sufficiently powered to detect the effect (a brief discussion of this point is included in lines 547-549).

4. In reporting their main results, the authors focus on how much longer infants looked at the fair than at the unfair event, but that seems to me an odd choice. What the authors find is that in many conditions (protector, victim, random), infants tended to show the typical effect, longer looking at the unfair than at the fair event. This reassures readers that the authors' method was generally sound as it yielded the typical effect. But in addition, the authors show us that in the bystander and aggressor conditions, there were deviations from this typical, expected pattern. From his perspective, it would make more sense to focus on looking at the unfair vs. the fair event, rather than that opposite. Figure 3, in particular, would be more intuitive to readers if it focused on the unfair event, as it says (incorrectly) on the vertical axis.

We agree that this framing would be more intuitive, and we have changed the reporting of these results throughout our Results section.

Other points:

76-80: "By the second year of life, infants expect agents to share resources fairly (i.e., equally) between two recipients (Schmidt & Sommerville, 2011; Sloane et al., 2012)". Actually, infants in the first year of life also expect agents to share resources equally between two recipients: see Meristo et al., 2016 (10 months), and Buyukozur Dawkins et al., 2019 (9 and 4 months). These studies support the claim that an expectation of fairness emerges early in infancy, and they should also be cited.

We have included citations to this work, but we have also noted that other scholars have challenged whether these findings reflect fairness expectations or social exclusion given that in these studies, one of the agents is not given any resources (Sommerville, 2024; lines 74-77).

101: the word infants is repeated.

Thank you for catching this error. We have removed the first instance of infants.

132-133: “In past studies, negatively valenced agents typically engaged in indirect harm towards recipients, such as hindering their goals (Gill & Sommerville, 2023), treating them unfairly (Surian et al., 2018)”. Is it possible that you accidentally reversed these two references?

We have reversed the order of these two citations (lines 96-99).

245-246: “Given that we are using the violation-of-expectation method, we also acknowledge that there is active debate over the validity of these methods (Aslin, 2007; Paulus, 2022).” It might be good to also cite the recent overview of the VOE paradigm by Margoni, Surian, and Baillargeon (2024), which presents the opposite side of this debate.

We have included a citation to Margoni et al. (2024) in this sentence (line 220)

252: “infants in the aggressor, protector, and victim conditions received identical test trials and highly similar test trials”. Identical familiarization trials?

Thank you for noticing this error, we have corrected it to read “identical familiarization trials and highly similar test trials” (line 225)

279-280: The video stopped once the aggressor contacted the protector, and a ding sound played to indicate that looking time coding should begin”. In experiments 1 and 2, the authors apparently coded infants’ looking behavior only during the paused scenes at the end of the trials. In their future research, however, the authors should consider also coding infants’ looking behavior during the long (here 32 s) event sequences that precede these paused scenes. It is important for the scientific community to know that infants in the different conditions attended about equally at the events shown in each trial. Moreover, infants who did not attend sufficiently to the events should be eliminated (VOE responses are meaningless if infants did not fully attend to the events).

Thank you for this suggestion. We will include this coding in future research.

298: perhaps “two novel recipients, green triangles”?

Thank you for this suggestion, we have added this to the manuscript (line 268).

318-320: “Further analyses are included in the Supplemental Materials to confirm that infants’ looking time in the familiarization trials did not significantly affect their looking time in the test trials (Supplemental Methods 2)”. There is something a bit odd about the familiarization data reported in the SI. In the cover letter, when comparing the familiarization data in the initial aggressor and protector conditions, the authors found a significant effect of condition. In the SI, when analyzing the familiarization data of the various conditions in Experiment 1, the authors again found an effect of condition, but now they write that “This difference in looking by condition was not particularly surprising given that infants saw different familiarization trials in the Bystander and Random Movement Conditions”. So it is unclear exactly what is going on here. Could the authors include a table with the mean familiarization trials in the 5 different conditions? Could they do follow-up analyses to inform their readers about which conditions reliably differed and which did not? And the same would apply to Experiment 2.

We have included the information suggested by the reviewer in the supplementary information (Supplementary Methods 2). In sum, in Experiment 1 infants looking towards the familiarization trials in the Protector condition differed from their looking towards the familiarization trials in the other conditions. Additionally, we have included the information on mean looking times as a graph rather than a table.

453: Infants in the Protector Condition?

We have corrected this error in line 417.

464: Ceiling babies are typically those who tend to look (near) the maximum allowed in both test events. We can see in Figure 4 that at least two infants in the aggressor condition of Experiment 2 looked near 30 s at both events; there were no such infants in the same condition of Experiment 1. Could including (as opposed to eliminating) those ceiling babies have contributed to the negative result of the aggressor condition in Experiment 2?

Removing infants who looked longer than 29.5 seconds (max 30 seconds) in either test trial did not change our results. Infants still significantly differentiated between each condition in Experiment 2 ($F(1,89) = 6.05, p = .016$), looked significantly longer towards the unfair test trial

in the Protector Condition ($t(46) = 2.61, p = .01$), and did not significantly differ in their looking in the Aggressor Condition ($t(43) = 1.05, p = .3$).

Similarly, these results also do not substantially differ if we only remove infants who looked longer than 29.5 seconds in *both* test trials.

537-540: “Infants’ capacity to make inferences concerning morally ambiguous agents also suggest that infants may view others’ character along a moral continuum rather than making categorical good vs bad judgements”. Ting and Baillargeon (2021) explicitly argue that infants evaluate moral character along a continuum (vicious, bad, good, virtuous) and present data that support such a notion.

We have cited Ting and Baillargeon (2021) in this sentence (lines 526-526).

545-546: “Our findings suggest that infants, at least in the current context, treat victims as virtuous”. In this literature, virtue often means going beyond what is morally required, like helping an outgroup member in need, or giving away a large share of a resource (see Ting & Baillargeon, 2021 for discussion). From this perspective, the victim was not virtuous, as noted above: It simply was assumed to possess a good character (there was no evidence to the contrary) and hence was expected to act fairly. In this scheme, infants in the present experiments assigned different characters to the agents: the aggressor was vicious (for infants who were surprised that it acted fairly), the bystander was bad (infants suspended their expectation that it would act fairly), the victim was good (infants expected it to act fairly), and the protector was presumably virtuous (there was no test of its virtue included in the experiments).

We agree that our wording here may have been misleading, so we have changed the wording in this paragraph from virtuous to “morally good” (line 513).

603-605: “By sampling a broad age range that included a range of social experience, the effect in Experiment 2 may have been diluted by increased variability among infants with lower social experience that are in this transitional period.”. This explanation for the discrepancy between the aggressor conditions in Experiments 1 and 2 is puzzling given that both experiments used the same age range, 12 to 24 months. Perhaps the authors mean that the larger sample used in Experiment 2 (48 vs. 32) introduced more variability in social experience?

We have clarified this point in line 569-576 by stating that the lower sample size in Experiment 1 may have led to increased sampling variability. This sampling variability may have resulted in differences in social experience between the two studies, which then may have led to differences in infants’ looking behaviours.